# Minimal Repairs for Learning Over Incomplete Data

## Abstract

Missing data often exists in real-world datasets, requiring significant time and effort for data repair to learn accurate machine learning (ML) models. In this paper, we show that imputing all missing values is not always necessary to achieve an accurate ML model. We introduce concepts of minimal and almost minimal repair, which are subsets of missing data items in training data whose imputation delivers accurate and reasonably accurate models, respectively. Imputing these sets can significantly reduce the time, computational resources, and manual effort required for learning models. We show that finding these sets is NP-hard for SVM and linear regression and propose efficient approximation algorithms with provable error bounds. Our extensive experiments indicate that our proposed algorithms can substantially reduce the time and effort required to learn on incomplete datasets.

## 1 Introduction

The performance of an ML model is highly dependent on the quality of its training data. In real-world data, a major data quality issue is missing or incomplete data Graham (2009); Kumar et al. (2017); Krishnan et al. (2016); Chu et al. (2016); Gupta et al. (2021). There are two common approaches to address missing values in training data. The first approach involves deleting samples with missing values. However, this method can lead to the loss of important information and introduce bias Van Buuren (2018). Another popular approach is data repair or imputation, in which end-users or ML practitioners impute missing values with the correct ones Andridge & Little (2010); Farhangfar et al. (2008); Le Morvan et al. (2021); Little & Rubin (2002); Śmieja et al. (2018); Pelckmans et al. (2005); Whang et al. (2023); Williams et al. (2005). Accurate repair is often challenging and expensive as it usually requires extensive collaboration with expensive domain experts. It usually must be repeated whenever the dataset evolves.

To reduce the cost of imputation, significant effort has been made to train imputation models on the observed subset of the dataset that predict accurate values for missing data items Buuren & Groothuis-Oudshoorn (2011); Kyono et al. (2021); Le Morvan et al. (2021); Stekhoven & Bühlmann (2011); Yoon et al. (2018); Zheng & Charoenphakdee (2022). State-of-the-art models for data imputation may take a long time to process and predict values for missing data items, and those that use deep neural networks need costly computational resources Perini & Nikolic (2024); Zheng & Charoenphakdee (2022); Yoon et al. (2018). As the dataset evolves, the user often has to repeat these steps. Moreover, in domains where important decisions must be made, e.g., healthcare and criminal justice, humans may need to manually verify the predictions of the imputation models Yakout et al. (2011). Some users also distrust black-box model-based imputation techniques in critical applications and prefer to reason about missing data themselves using observed features and domain knowledge Ahmad et al. (2019); Stempfle et al. (2025). In addition, model-based imputation may perform poorly when the ratio of missing data to observed data is too large Dettori JR (2018); Jakobsen et al. (2017); Junaid et al. (2025). In these settings, users may have to manually repair at least parts of the data.

To address these challenges, we introduce the concept of a **minimal repair** for a training dataset with missing values. Generally speaking, this set represents the smallest group of data items with missing values that, once repaired, yields the same model as that trained on a fully and accurately repaired dataset. By finding and imputing this set, users can significantly reduce the time and effort required to manually repair a dataset without sacrificing model accuracy. It also reduces the time and computational resources needed to predict missing values using imputation models and the manual

labor required to verify their imputations. Moreover, minimal repair of a dataset pinpoints the subset of the dataset whose uncertainty impacts the effectiveness of the model trained on the dataset. Hence, it simplifies the inspection and debugging of model training, which is often labor intensive Siddiqi et al. (2023). Because incomplete data sets are prevalent and often evolve, a small reduction in time, effort, and computational resources in the preparation of training datasets can save significant resources in the long run. Specifically, our contributions are as follows.

- We define minimal repair for learning support vector machines (SVM) (Section 3) and linear regression (Section 4) over incomplete data. We prove that finding minimal repairs for SVM and linear regression is NP-hard and propose efficient algorithms with provable error bounds to approximate minimal repairs for them.

- Minimal repairs may sometimes be too large or take too long to find. We propose the concept of **almost minimal repair**, which is the minimal subset of data items with missing values whose repair delivers a model with a loss within a given threshold from the model trained over the fully and accurately repaired dataset. We prove that the problem of finding almost minimal repairs is NP-hard for SVM and linear regression and propose algorithms with provable error bounds to approximate almost minimal repairs for them (Section 5).

- We evaluated the scalability of our algorithms on multiple real-world datasets (Section 6). Our empirical results indicate that our proposed algorithms efficiently approximate minimal and almost minimal repairs and deliver models with the same or almost the same accuracy as those trained over fully repaired datasets. Our results also indicate that using minimal and almost minimal repairs can reduce the time of the model-based imputation methods for large data without losing accuracy in the downstream learning task.

## 2 BACKGROUND

We model the training data as a table where each row represents a training sample. One column in the table represents labels and others represent the features of the samples. Given that the training data has $d$ features, we denote its features as $[\mathbf{z}_1, \ldots, \mathbf{z}_d]$. The values of each feature belong to the *domain of the feature*, e.g., real numbers. To simplify our analysis, we assume that all the features share the same domain. Our results extend to other settings. A *training set* with $n$ samples is a pair of a feature matrix $\mathbf{X} = [\mathbf{x}_1, ..., \mathbf{x}_n]^T$ and a corresponding label vector $\mathbf{y} = [y_1, ..., y_n]^T$. We denote each sample with $d$ features in $\mathbf{X}$ as a vector $\mathbf{x}_i = [x_{i1}, ..., x_{id}]$, where $x_{ij}$ represents the $j^{th}$ feature in the $i^{th}$ sample. Given the training set $(\mathbf{X}, \mathbf{y})$, the target function $f$, and the loss function $L$, the goal of training is to find an optimal model $\mathbf{w}^* = \arg\min_{\mathbf{w} \in \mathcal{W}} L(f(\mathbf{X}, \mathbf{w}), \mathbf{y})$.

**Missing values** Any $x_{ij}$ is a missing value if it is unknown (marked by *null*). An *incomplete sample* (*incomplete feature*) is a sample (feature) with at least one missing value. We use *complete feature* and *complete sample* to refer to features and samples that are free of missing values. We denote the set of all missing values in a feature matrix $\mathbf{X}$ as $M(\mathbf{X})$, the set of incomplete samples as $MS(\mathbf{X})$, and the set of incomplete features as $MF(\mathbf{X})$. In this paper, we focus on the case where all missing values are in the feature matrix and the label vector is complete.

**Repair** A repair is a complete version of an incomplete feature matrix $\mathbf{X}$ where all missing values in $\mathbf{X}$ are replaced with values from their domains and the complete values of $\mathbf{X}$ remain intact. Given the repair $\mathbf{X}^r$ of the feature matrix $\mathbf{X}$, we denote the repair, i.e. imputation, of the sample $\mathbf{x}_i$ in $\mathbf{X}$ by $\mathbf{x}_i^r$. Since the domains of features often contain numerous or infinite values, an incomplete feature matrix usually has many or infinitely many repairs. We denote this set of all repairs of $\mathbf{X}$ by $\mathbf{X}^R$.

## 3 MINIMAL REPAIR (MR) FOR SVM

We use the concept of certain model Zhen et al. (2024) to define minimal repair for SVM. A model $\mathbf{w}^*$ is a certain model for the target function $f$ on the training set $(\mathbf{X}, \mathbf{y})$ if for every repair $\mathbf{X}^r \in \mathbf{X}^R$, we have $\mathbf{w}^* = \arg\min_{\mathbf{w} \in \mathcal{W}} L(f(\mathbf{X}^r, \mathbf{w}), \mathbf{y})$ where $L$ is the loss function. Intuitively, a certain model minimizes training loss for all repairs of the incomplete feature matrix. Thus, if a certain model exists,

one can learn an accurate model over the training set without any repair to the training data, as training over any repair to the dataset, e.g., using randomly selected values, will deliver the same accurate model. This observation holds regardless of the missingness mechanism—Missing Completely at Random (MCAR), Missing at Random (MAR), or Missing Not at Random (MNAR). Given the restrictive definition of certain models, they do not often exist Zhen et al. (2024). Thus, we find the minimal repair of an incomplete training set such that the resulting training set has a certain model.

**Definition 1** *A set of incomplete samples $\mathbf{S}_{MR}$ in the training set $(\mathbf{X}, \mathbf{y})$ is a minimal repair for learning SVM with the regularization parameter $C$ if we have: 1) a certain model exists when imputing all missing values in $\mathbf{S}_{MR}$, and 2) there is no other set $\mathbf{S}'$ satisfying condition (1) such that $|\mathbf{S}'| < |\mathbf{S}_{MR}|$ where $|\mathbf{S}_{MR}|$ denotes the cardinality of $\mathbf{S}_{MR}$.*

According to Definition 1, the certain model exists regardless of the imputed values in $\mathbf{S}_{MR}$. Obviously, if these values are accurate, e.g., by using experts or effective imputation models, or inaccurate, the certain model will be accurate or inaccurate, respectively. Our aim is *not* to improve the accuracy of imputations directly, but to reduce resources used for such imputations using minimal repairs.

We denote the minimal repair for SVM with the regularization parameter $C$ on the training set $(\mathbf{X}, \mathbf{y})$ as $\mathbf{S}_{MR}(\mathbf{X}, \mathbf{y}, C)$. We have the following property for the minimal repair of SVM.

**Theorem 1** *Given training set $(\mathbf{X}, \mathbf{y})$ and regularization parameter $C$, $\mathbf{S}_{MR}(\mathbf{X}, \mathbf{y}, C)$ is unique.*

### 3.1 Finding Minimal Repair

Let $SV(\mathbf{X^r}, \mathbf{y}, C)$ be the set of support vectors for the optimal SVM model with regularization parameter $C$ on a repair $\mathbf{X^r}$ of the training set $(\mathbf{X}, \mathbf{y})$.

**Lemma 2** *Given the training set $(\mathbf{X}, \mathbf{y})$ and the regularization parameter $C$, at least one repair $\mathbf{x}_i^r$ of every sample $\mathbf{x}_i \in S_{MR}(\mathbf{X}, \mathbf{y}, C)$ is a support vector in a repair $\mathbf{X^r}$ of $\mathbf{X}$, i.e., $\mathbf{x}_i^r \in SV(\mathbf{X^r}, \mathbf{y}, C)$.*

Hence, to determine if an incomplete sample belongs to the minimal repair, one could materialize every repair of the feature matrix and check if the incomplete sample is a support vector for any of them. However, this process can be extremely inefficient due to the often large number of repairs. Assume that each missing value $x_{ij}$ is bounded by an interval $[x_{ij}^{min}, x_{ij}^{max}]$ based on its domain. $\mathbf{X}^e$ is an *edge repair* to $\mathbf{X}$ if for every missing value $x_{ij}$, $x_{ij}^e = x_{ij}^{min}$ or $x_{ij}^{max}$. $\mathbf{X}^E$ denotes the set of all edge repairs for $\mathbf{X}$. Theorem 3 shows that we can use only the edge repair instead of all repairs to check if an incomplete sample belongs to the minimal repair.

**Theorem 3** *Given the training set $(\mathbf{X}, \mathbf{y})$ and the regularization parameter $C$, an incomplete sample $\mathbf{x}_i$ belongs to minimal repair $S_{MR}(\mathbf{X}, \mathbf{y}, C)$ if and only if there is at least one edge repair $\mathbf{X}^e$ of $\mathbf{X}$ such that $\mathbf{x}_i^e \in SV(\mathbf{X}^e, \mathbf{y}, C)$ where $\mathbf{x}_i^e$ is the repair of $\mathbf{x}_i$.*

Based on Theorem 3, we can find the minimal repair following these steps: 1) Initialize an empty minimal repair set, $S_{MR}$. 2) Iterate over each incomplete sample $\mathbf{x}_i$. At each iteration, materialize all edge repairs $\mathbf{X}^e \in \mathbf{X}^E$, and check if $\mathbf{x}_i$ is a support vector for any of the edge repairs. If it is, add $\mathbf{x}_i$ to $S_{MR}$, and 3) Finally, return the minimal repair $S_{MR}$. Despite this optimization, finding the minimal repair remains computationally intractable.

**Theorem 4** *Given a training set $(\mathbf{X}, \mathbf{y})$ with missing values, deciding whether an incomplete sample belongs to the minimal repair for SVM on $(\mathbf{X}, \mathbf{y})$ is NP-hard. Consequently, finding the minimal repair for SVM on $(\mathbf{X}, \mathbf{y})$ is NP-hard.*

### 3.2 Approximating Minimal Repair

We propose an efficient approximation algorithm (Algorithm 1) to find minimal repair for SVM. Its key idea is to test whether each incomplete sample $\mathbf{x}_i$ belongs to minimal repair by constructing an edge repair $\mathbf{X}^e$ that maximizes the likelihood of $\mathbf{x}_i$ becoming a support vector. This construction begins with a random edge repair and iteratively updates each missing value in the dataset to its minimum or maximum bound. At each step, this choice minimizes $y_i \mathbf{w}^\top \mathbf{x}_i$, encouraging $\mathbf{x}_i$ to

satisfy the support vector condition $y_i \mathbf{w}^\top \mathbf{x}_i \leq 1$. If this condition holds after the full pass of the data, $\mathbf{x}_i$ is selected for repair. Crucially, this algorithm **does not return any false positive**. Since the algorithm initializes with a randomly selected edge repair, it does not introduce bias towards any specific imputation in learning models.

**Theorem 5** *Every sample returned by Algorithm 1 belongs to $S_{MR}(\mathbf{X}, \mathbf{y}, C)$.*

---

**Algorithm 1** Approximating minimal repair for SVM on training set $(\mathbf{X}, \mathbf{y})$

---

$S_{MR} \leftarrow [\quad]$
$\mathbf{X}^e \leftarrow$ a random edge repair to the feature matrix $\mathbf{X}$
**for** $\mathbf{x}_i \in MS(\mathbf{X})$ **do**
    **for** $x_{pq} \in M(\mathbf{X})$ **do**
        $\mathbf{X}^{e_{min}}, \mathbf{X}^{e_{max}} \leftarrow$ two edge repairs by only replacing $x_{pq}$ in $\mathbf{X}^e$ with its min or max value
        $\mathbf{w}_1, \mathbf{w}_2 \leftarrow SVM(\mathbf{X}^{e_{min}}, \mathbf{y}), SVM(\mathbf{X}^{e_{max}}, \mathbf{y})$ {learning SVM models with edge repairs}
        $\mathbf{X}^e \leftarrow$ **if** $y_i \mathbf{w}_1^\top \mathbf{x}_i^{e_{min}} \leq y_i \mathbf{w}_2^\top \mathbf{x}_i^{e_{max}}$ **then** $\mathbf{X}^{e_{min}}$ **else** $\mathbf{X}^{e_{max}}$
    **end for**
    $\mathbf{w} \leftarrow SVM(\mathbf{X}^e, \mathbf{y})$
    **if** $y_i \mathbf{w}^\top \mathbf{x}_i \leq 1$ **then** $S_{MR} \leftarrow S_{MR}.add(\mathbf{x}_i)$
**end for**
return $S_{MR}$

---

Since each iteration modifies only one missing value, adjacent models $\mathbf{w}_1$ and $\mathbf{w}_2$ differ by only a single feature entry. This allows us to avoid retraining from scratch by applying incremental or decremental SVM updates Cauwenberghs & Poggio (2000); Laskov et al. (2006). These techniques update the model efficiently—typically an order of magnitude faster—by reusing computations from the previous solution.

Algorithm 1 may miss some samples of minimal repair. Thus, we iteratively apply Algorithm 1 to the remaining incomplete samples in the training set to find more samples in the minimal repair of the training set. The process ends when no new samples are selected for repair. The following theorem shows that the probability of not finding samples of minimal repair decreases using this approach.

**Theorem 6** *Given the training set $(\mathbf{X}, \mathbf{y})$, let $p_k(\mathbf{x})$ be the probability that an incomplete sample $\mathbf{x}$ in minimal repair of $(\mathbf{X}, \mathbf{y})$ not returned in iteration of $k > 0$ in iterative application of Algorithm 1, $p_k(\mathbf{x}) > p_{k+1}(\mathbf{x})$.*

**Corollary 6.1** *If the probability distribution of each missing value is known, and we let $g(x_{ij})$ denote the probability density function of the ground truth value for the missing value $x_{ij}$ in the incomplete training set $(\mathbf{X}, \mathbf{y})$. If missing values in $\mathbf{X}$ are independent, the probability that an incomplete sample $\mathbf{x}_i$ in minimal repair not returned by Algorithm 1 in the main content is:*

$$p(\mathbf{x}_i) = 1 - \frac{\int \cdots \int_{\min(x_{ij}^{visited})}^{\max(x_{ij}^{visited})} \prod_{x_{ij} \in M(\mathbf{X})} g(x_{ij}) \, dx_{ij}}{\int \cdots \int_{x_{ij} \in M(\mathbf{X})} \prod_{x_{ij} \in M(\mathbf{X})} g(x_{ij}) \, dx_{ij}} \tag{1}$$

$x_{ij}^{visited} \in \{x_{ij}^{min}, x_{ij}^{max}\}$ *shows the values used for $x_{ij}$ in Algorithm 1.*

## 4 MINIMAL REPAIR FOR LINEAR REGRESSION

The minimal repair for linear regression is the smallest set of features that is necessary to repair.

**Definition 2** *Given the training set $(\mathbf{X}, \mathbf{y})$, a set of incomplete features in $\mathbf{X}$, denoted as $\mathbf{S}_{MR}(\mathbf{X}, \mathbf{y})$, is a minimal repair for $(\mathbf{X}, \mathbf{y})$ for linear regression if we have: 1) a certain model exists upon imputing all missing values in the $\mathbf{S}_{MR}(\mathbf{X}, \mathbf{y})$, and 2) there is no set $\mathbf{S}$ satisfying condition (1) and $|\mathbf{S}| < |\mathbf{S}_{MR}|$.*

Similarly to Definition 1, the existence of certain models in Definition 2 is orthogonal to the accuracy of imputation to $\mathbf{S}_{MR}$. In linear regression, the optimal linear regression model $\mathbf{w}^*$ consists of the

set of linear coefficients for feature vectors. A feature $z_i$ is considered relevant if the corresponding linear coefficient in the optimal model $w_i^*$ is not zero, and it is irrelevant if $w_i^*$ equals zero. Intuitively, an incomplete feature needs to be repaired if it is relevant (i.e., it plays a role in the optimal model) and does not need to be repaired if it is irrelevant. However, traditional statistical tools, such as the chi-square test, require complete distributions for each feature to assess correlations, which is challenging in the presence of missing values. The minimal repair may not be unique.

**Theorem 7** *There is a training set with multiple minimal repairs for linear regression. In addition, if all the features in all the repairs of the training set $(\mathbf{X}, \mathbf{y})$ are linearly independent, the minimal repair for linear regression over $(\mathbf{X}, \mathbf{y})$ is unique.*

The following theorem establishes that finding minimal repair for linear regression is intractable.

**Theorem 8** *Given a training set $(\mathbf{X}, \mathbf{y})$ with incomplete features, finding the minimal repair for linear regression over $(\mathbf{X}, \mathbf{y})$ is NP-hard.*

To find minimal repair efficiently, we first propose an equivalent problem in Theorem 9, based on a variant of the well-known sparse linear regression problem Bruckstein et al. (2009).

**Lemma 9** *Finding the minimal repair for linear regression on training set $(\mathbf{X}, \mathbf{y})$ is equivalent to:*

$$
\begin{aligned}
&\min_{\mathbf{w} \in \mathcal{W}} T_{MF(\mathbf{X})}(\mathbf{w}) \\
&subject\ to \quad \mathbf{w} = \arg\min ||\mathbf{X}^r \mathbf{w} - \mathbf{y}||_2^2, \forall \mathbf{X}^r \in \mathbf{X}^R
\end{aligned}
\tag{2}
$$

*where $T_{MF(\mathbf{X})}(\mathbf{w})$ is the number of non-zero linear coefficient in $\mathbf{w}$ whose corresponding feature is incomplete, i.e., $T_{MF(\mathbf{X})}(\mathbf{w}) = |\{\mathbf{z}_i \in MF(\mathbf{X})|w_i! = 0\}|$*

The distinction between our problem and sparse linear regression lies in their objectives: sparse linear regression seeks to minimize the number of non-zero coefficients across all features, whereas we focus on minimizing the number of non-zero coefficients only among incomplete features. Orthogonal Matching Pursuit (OMP) provides an efficient approximation to solve the sparse linear regression problem Wang et al. (2012). This greedy algorithm begins with an empty solution set and initializes the regression residual to the label vector. In each iteration, the algorithm selects the feature most relevant to the current residual (with largest dot product), adds it to the solution set, retrains a linear regression model, and updates the residual accordingly. It stops when the regression residue is sufficiently small.

We propose a variant of OMP, as outlined in the appendix, to find minimal repair for linear regression. Our algorithm has two major differences compared to the conventional OMP. First, we include all complete features in the regression at the initialization, ensuring that we minimize the number of non-zero coefficients only among incomplete features. Secondly, we define our stopping condition by the maximum relevance (cosine similarity) between the feature and the label being smaller than or equal to a user-defined threshold, instead of relying on a near-zero regression residue. This approach enables our algorithm to work with general datasets without requiring the assumption of an underdetermined linear system, which is typically necessary in conventional OMP.

The time complexity of the algorithm is $\mathcal{O}(T_{train} \cdot |MF(\mathbf{z})|)$, making it significantly more efficient than the baseline algorithm, which trains models on all repairs individually and has a time complexity of $\mathcal{O}(T_{train} \cdot |\mathbf{X}^R|)$. If we use gradient descent, our algorithm has a time complexity of $\mathcal{O}(n \cdot d^3)$, where $n$ is the number of training samples and $d$ is the number of features. In cases where $n < d^2$, the time complexity is reduced to $\mathcal{O}(n \cdot d^2 + n^2 \cdot d)$ under certain conditions by applying incremental learning techniques based on the Sherman-Morrison formula, as outlined in the appendix. The following theorem characterizes the approximation rate of our algorithm.

**Theorem 10** *The first $k$ incomplete features added to $S_{MR}$ in our algorithm for training set $(\mathbf{X}, \mathbf{y})$ belong to a minimal repair of $(\mathbf{X}, \mathbf{y})$ with a probability of at least $1 - 1/n$, provided that: 1) $\mu < 1/(2k-1)$, 2) the missing values in the dataset follow independent zero-mean normal distributions ($\mathcal{N}(0, \sigma_{ij}^2)$), and 3) all linear coefficients ($w_i, \mathbf{z}_i \in MF(\mathbf{X})$) for incomplete features satisfy:*

$$
|w_i| \geq \frac{2 \sum_{x_{ij}=null} \sigma_{ij} \sqrt{n + 2\sqrt{n \log n}}}{1 - (2k-1)\mu}
\tag{3}
$$

*where $\mu$ is the mutual incoherence defined by $\mu = \max\limits_{i \neq j} |\mathbf{z_i}^T \mathbf{z_j}|$.*

## 5 ALMOST MINIMAL REPAIR

Minimal repair might be too large and take a long time to compute for some datasets and learning tasks. Thus, we relax the definition of minimal repair to reduce its size and computation cost. Instead of enforcing exact optimality, we aim for a set whose imputation can deliver a model that is near-optimal for all possible repairs. We use the concept of approximately certain model (ACM) Zhen et al. (2024) to formalize this notion. For a user-defined threshold $e \geq 0$, $\mathbf{w}^{\approx}$ is an ACM for target function $f$ on training set $(\mathbf{X}, \mathbf{y})$ if for every repair $\mathbf{X}^r$, $L(\mathbf{w}^{\approx}, \mathbf{X}^r, \mathbf{y}) - \min_{\mathbf{w} \in \mathcal{W}} L(\mathbf{w}, \mathbf{X}^r, \mathbf{y}) \leq e$.

**Definition 3** *Given a threshold $e \geq 0$, a set $S_{AMR}$ of incomplete samples in the training set $(\mathbf{X}, \mathbf{y})$ is an almost minimal repair (AMR) for the target function $f$ with loss $L$ if: (1) repairing $S_{AMR}$ yields an ACM for $f$ in $(\mathbf{X}, \mathbf{y})$, and (2) no other set $S'$ satisfies (1) with $|S'| < |S_{AMR}|$.*

If $e = 0$, ACM reduces to a certain model. Hence, we can show that computing AMR is also NP-hard for SVM (details are in the appendix).

### 5.1 COMPUTING AMR

We first propose an iterative algorithm with two main steps. Step 1 (ST1: ACM Optimizer) takes the input dataset in iteration $k > 0$ of the algorithm, $\mathbf{X}^{(k)}$, and finds the model $\mathbf{w}_k^{\approx}$ that minimizes the worst-case suboptimality gap $g_k = \sup_{\mathbf{X}^{(k)r}} \left[ L(\mathbf{w}_k^{\approx}, \mathbf{X}^{(k)r}, \mathbf{y}) - \min_{\mathbf{w} \in \mathcal{W}} L(\mathbf{w}, \mathbf{X}^{(k)r}, \mathbf{y}) \right]$.

Step 2 (ST2: Local Repair Set Identifier) examines whether $g_k > e$, and if so, returns the smallest set of currently incomplete samples whose imputation may help further reduce the suboptimality gap in the next iteration.

**Theorem 11** *Given the training set $(\mathbf{X}, \mathbf{y})$, each selection made by ST2 belongs to the AMR set $S_{AMR}$ of $(\mathbf{X}, \mathbf{y})$. Thus, the iterative algorithm terminates with an ACM, and the total imputed set $S_{iter\text{-}ACM} \subseteq S_{AMR}$, where $S_{iter\text{-}ACM}$ is the union of all incomplete samples selected across iterations.*

This guarantees that our algorithm converges to an ACM by imputing only a subset of $S_{AMR}$. The key distinction is that $S_{AMR}$ is defined to guarantee the ACM condition under all possible repairs—it is sufficient without knowledge of any imputation results. In contrast, the iterative algorithm dynamically learns imputation results along the way. This new information may render some samples in $S_{AMR}$ unnecessary for achieving ACM in the current trajectory. Thus, $S_{iter\text{-}ACM}$ can be smaller than $S_{AMR}$ while still ensuring the ACM condition.

### 5.2 EFFICIENT APPROXIMATION

Both ST1 and ST2 are intractable because they require solving min-sup optimization over exponentially many repairs and identifying minimal subsets of incomplete samples whose repair is necessary when an ACM does not yet exist. Specifically, these are the samples whose imputation would further reduce the minimum value of the worst-case suboptimality gap $g(\mathbf{w}) = \sup_{\mathbf{X}^r} h(\mathbf{w}, \mathbf{X}^r)$ toward the user-defined threshold $e$. Finding such subsets involves understanding how each missing value affects the supremum over all repairs—a problem known to be computationally hard in general due to the nested structure of min-max optimization Ben-Tal et al. (2008). We therefore propose efficient approximations of these steps that make the entire algorithm tractable.

**Approximating ST1 (ACM Optimizer):** ST1 aims to find the model $\mathbf{w}_k^{\approx} = \arg\min_{\mathbf{w}} \sup_{\mathbf{X}^r \in \mathcal{X}_{\text{rem}}^R} h(\mathbf{w}, \mathbf{X}^r)$, where $h(\mathbf{w}, \mathbf{X}^r) = L(\mathbf{w}, \mathbf{X}^r) - \min_{\mathbf{w}'} L(\mathbf{w}', \mathbf{X}^r)$. When the loss function $L$ is convex, each $h(\mathbf{w}, \mathbf{X}^r)$ is convex in $\mathbf{w}$, and so is the pointwise supremum of such functions. Thus, we approximate this by sampling a finite subset of edge repairs $\{\mathbf{X}_1^e, \ldots, \mathbf{X}_s^e\}$ and solving the convex problem $\min_{\mathbf{w}} \max_i h(\mathbf{w}, \mathbf{X}_i^e)$.

However, directly computing $h(\mathbf{w}, \mathbf{X}^e)$ requires solving an inner optimization for each sampled repair to obtain the minimum loss. To make this tractable, we use the subgradient norm $\|\mathbf{g}(\mathbf{w}, \mathbf{X}^e)\|$ as a proxy for the suboptimality gap.

Table 1: Details of datasets with injected missing data

| Data Set | Task | Features | Training samples | Missing Factor% | Missingness Type |
|---|---|---|---|---|---|
| Malware | Classification | 6823 | 1596 | 20-40-60 | MCAR, MAR, MNAR |
| Tuadromd | Classification | 242 | 3571 | 20-40-60 | MCAR, MAR, MNAR |
| Credit Default | Classification | 23 | 30000 | 20-40-60 | MCAR, MAR, MNAR |
| Gas | Regression | 129 | 2566 | 20-40-60 | MCAR |
| Superconductivity | Regression | 82 | 21262 | 20-40-60 | MCAR |
| Concrete | Regression | 8 | 1030 | 20-40-60 | MCAR |

Table 2: Details of datasets with original missing data

| Data Set | Task | Features | Training samples | Missing Factor | Missingness Type |
|---|---|---|---|---|---|
| Breast Cancer | Classification | 10 | 559 | 1.97% | MCAR |
| Water-Potability | Classification | 9 | 2620 | 39.00% | MCAR |
| Online-Ed | Classification | 36 | 7026 | 35.48% | MNAR, MCAR |
| Bankruptcy | Classification | 64 | 8402 | 54.00% | MNAR |
| Air Quality | Regression | 12 | 7344 | 90.80% | MNAR |
| Communities | Regression | 1954 | 1595 | 93.67% | MCAR |
| Cancer Rate | Regression | 32 | 3048 | 81.00% | MCAR |

**Theorem 12** *If $L(\mathbf{w})$ is convex and has an $M$-Lipschitz continuous gradient, then any model $\mathbf{w}^{\approx}$ satisfying $\|\nabla_{\mathbf{w}} L(\mathbf{w}, \mathbf{X}^r)\| \leq \sqrt{2Me}$ for all $\mathbf{X}^r$ is an ACM.*

This result implies that for linear regression, which satisfies the convexity and smoothness conditions, we can directly use the gradient norm to check whether a model is an ACM. For non-differentiable models like linear SVM, the hinge loss is not smooth and the subgradient norm is not convex. Nonetheless, we still use the subgradient norm as a practical stopping proxy to assess whether ACM has been achieved.

**Approximating ST2 (Local Repair Set Identifier):**   ST2 must find a small subset of currently incomplete samples whose repair enables further progress toward satisfying the ACM condition. We approximate this by identifying edge repairs $\mathbf{X}^e$ from the sampled set where $\|\mathbf{g}(\mathbf{w}_{\widetilde{k}}^{\approx}, \mathbf{X}^e)\| > \epsilon'$, indicating that ACM is violated under these repairs.

We then inspect each such "problematic" edge repair. For each incomplete sample $x_j$ that currently violates the margin condition (i.e., $y_j(\mathbf{w}_{\widetilde{k}}^{\approx})^T \mathbf{x}_j^e < 1$), we check if there exists a feasible repair where the margin would exceed 1. If so, we assign a score to $x_j$ estimating its potential to reduce the subgradient norm. One option is the maximum hinge loss reduction:

$$\Delta L_{\max} = C \cdot \left[(1 - \text{margin}_j) - \max(0, 1 - \text{margin}_{j,\max})\right], \tag{4}$$

where $\text{margin}_{j,\max}$ is estimated using interval arithmetic over the missing feature bounds. Alternatively, we compute a gradient alignment score based on the inner product between the current subgradient vector and $Cy_j\mathbf{x}_j^e$, estimating the contribution to gradient magnitude.

These scores are aggregated across all high-gradient edge repairs. We then select the top-$h$ highest-ranked incomplete samples for imputation in the next iteration. This procedure effectively approximates the function of ST2, enabling tractable, targeted refinement of the model toward satisfying the ACM condition.

## 6 EXPERIMENTAL EVALUATION

We have evaluated our methods on six real-world datasets with injected missingness and seven with naturally occurring missing values, spanning diverse domains and varying in missingness ratios (proportion of incomplete samples), feature dimensionalities, sample sizes, and types of missingness (Tables 1 and 2). Details on datasets and the experiment setting are in the appendix.

As explained in Section 1, users manually repair their data in some settings. Thus, we compare the accuracy and time overhead of our methods to *Active Clean (AC)* Krishnan et al. (2016), which

Table 3: Accuracy/time for SVM on data with injected MCAR

| Data Set | % Missing | Ground Truth Accuracy(%) | Time(s) | | | Accuracy(%) | | | Impute % of Samples | | |
|---|---|---|---|---|---|---|---|---|---|---|---|
| | | | AC | MR | AMR | AC | MR | AMR | AC | MR | AMR |
| Malware | 20 | 95.61 | **1.36** | 6.15 | 14.5 | 93.13 | **96.7** | 95.3 | 6.39 | 18.68 | **1.49** |
| | 40 | 95.03 | **0.56** | 98.0 | 14.9 | 92.20 | 92.42 | **95.1** | 3.35 | 21.1 | **0.69** |
| | 60 | 95.91 | **0.170** | 12.81 | 15.1 | 88.67 | **96.37** | 94.8 | 3.28 | 16.65 | **0.46** |
| Tuadromd | 20 | 98.67 | **0.68** | 1.135 | 2.7 | 97.53 | **98.73** | 98.2 | 3.78 | 11.9 | **1.3** |
| | 40 | 98.77 | **0.54** | 2.19 | 2.9 | 97.42 | **98.81** | 98.2 | 3.53 | 11.1 | **0.7** |
| | 60 | 98.77 | **0.34** | 3.29 | 2.8 | 97.50 | **98.77** | 98.2 | 2.48 | 11.8 | **0.4** |
| Credit Default | 20 | 81.03 | 11.86 | **1.39** | 3.4 | **81.02** | **81.02** | 77.8 | 0.19 | 30 | **0.05** |
| | 40 | 81.03 | 14.19 | 3.93 | **3.9** | **81.02** | 81.00 | 77.7 | 0.23 | 30 | **0.03** |
| | 60 | 81.02 | 14.2 | **0.57** | 3.7 | **81.02** | **81.02** | 77.8 | 0.19 | 30 | **0.01** |

Table 4: Accuracy/time for SVM on data with original missingness using model-based imputation

| Data Set | Method | Time(s) | | | | Accuracy(%) | | | | % Samples Imputed |
|---|---|---|---|---|---|---|---|---|---|---|
| | | KNN | MICE | TCSDI | MF | KNN | MICE | TCSDI | MF | |
| Breast Cancer | MR | 0.055 | 0.064 | 51 | 0.124 | 96.30 | 96.4 | 97.00 | 96.58 | 18.2 |
| | AMR | 0.016 | **0.019** | 3.83 | 0.127 | 96.43 | 96.78 | 97.11 | 96.87 | **1.37** |
| | AC | 0.065 | 0.065 | 84 | **0.120** | 95.85 | 96.30 | **97.87** | 96.80 | 87.27 |
| | Baseline | **0.0039** | 0.046 | 102 | 2.80 | 95.78 | 96.30 | 97.00 | 97.0 | 100 |
| Water-Potability | MR | 0.259 | 0.135 | 473.7 | 9.91 | 60.2 | 60.3 | **62.80** | 60.1 | 30 |
| | AMR | 2.03 | 1.95 | **10.51** | **2.22** | **60.98** | **60.98** | 60.13 | **60.98** | **0.72** |
| | AC | 0.33 | 0.033 | 85.32 | 5.47 | 54.96 | 56.90 | 57.00 | 58.19 | 1.94 |
| | Baseline | **0.0053** | **0.0115** | 1459 | 12.8 | 60.53 | 60.63 | 61.3 | 60.53 | 100 |
| Online-Ed | MR | 1.606 | **0.748** | 1087.2 | 8.31 | 64.5 | 64.5 | 65.22 | 63.78 | 29.91 |
| | AMR | 3.45 | 3.56 | **15.45** | **3.83** | 63.86 | 62.70 | 63.87 | 63.86 | **0.43** |
| | AC | 1.83 | 1.88 | 93.76 | 6.23 | 63.71 | 60.77 | 63.60 | 63.41 | 0.81 |
| | Baseline | **0.989** | 1.270 | 3624 | 17.09 | **65.23** | 65.17 | **65.23** | **65.23** | 100 |
| Bankruptcy | MR | 2.798 | **0.76** | 2286.7 | 451.3 | 97.22 | 97.8 | 97.79 | 96.04 | 29.9 |
| | AMR | 10.85 | 15.89 | **23.71** | **25.59** | 95.29 | 96.40 | 97.11 | 95.29 | **0.31** |
| | AC | **2.24** | 2.25 | 101 | 250.3 | 96.01 | 96.41 | 96.78 | 96.52 | 0.6 |
| | Baseline | 4.843 | 22.15 | 7620 | 710.16 | 96.00 | 96.30 | 97.00 | **97.46** | 100 |

Table 5: Accuracy/time for Linear Regression on data with injected MCAR

| Data Set | % Missing | Ground Truth MSE | Time(s) | | | MSE | | | Impute % of Samples or Features | | |
|---|---|---|---|---|---|---|---|---|---|---|---|
| | | | AC | MR | AMR | AC | MR | AMR | AC | MR | AMR |
| Superconductivity | 20 | 0.0088 | **2.20** | 2.305 | 3.17 | **0.0884** | 0.00888 | 0.105 | 0.24 | 70.00 | **0.05** |
| | 40 | 0.0088 | 2.23 | 2.534 | 3.24 | 0.00886 | **0.00885** | 0.102 | 0.22 | 75.00 | **0.03** |
| | 60 | 0.0088 | **1.46** | 2.476 | 3.15 | 0.089 | **0.00885** | 0.102 | 0.25 | 75.00 | **0.01** |
| Gas | 20 | 0.1053 | **0.0734** | 0.31 | 1.16 | 0.114 | **0.105** | 0.161 | 2.01 | 65.00 | **0.02** |
| | 40 | 0.1053 | 0.051 | 0.3391 | 1.24 | 0.112 | **0.1054** | 0.160 | 2.01 | 65.00 | **0.01** |
| | 60 | 0.1053 | **0.0332** | 0.551 | 1.15 | 0.117 | **0.112** | 0.157 | 1.78 | 25.00 | **0.01** |
| Concrete | 20 | 0.0149 | **0.0126** | 0.0227 | 0.3432 | 0.0152 | **0.01495** | 0.0541 | 6.89 | 50.00 | **0.07** |
| | 40 | 0.0149 | **0.0149** | 0.0202 | 0.3587 | 0.0151 | **0.01495** | 0.0541 | 5.63 | 50.00 | **0.04** |
| | 60 | 0.0149 | **0.0065** | 0.0199 | 0.3011 | 0.0156 | **0.01495** | 0.0541 | 5.28 | 50.00 | **0.02** |

Table 6: Accuracy/time for Linear Regression on data with original missing using model-based imputation

| Data Set | Method | Time(s) | | | | MSE | | | | % Samples Imputed |
|---|---|---|---|---|---|---|---|---|---|---|
| | | KNN | MICE | TCSDI | MF | KNN | MICE | TCSDI | MF | |
| Cancer Rate | MR | **0.153** | 0.574 | 5852 | 6.89 | **0.0045** | **0.0045** | **0.0045** | **0.0045** | 33.3 |
| | AMR | 0.35 | 1.17 | 45.74 | **0.78** | 0.0047 | 0.0045 | **0.0045** | **0.0045** | **0.48** |
| | AC | 0.166 | **0.133** | **110** | 3.95 | 0.0050 | 0.0051 | 0.0049 | 0.0049 | 0.70 |
| | Baseline | 0.584 | 0.664 | 6104 | 7.24 | **0.0045** | 0.0058 | 0.0049 | 0.0047 | 100 |
| Air Quality | MR | 1.06 | 2.46 | 14976 | 2.24 | **5.671** | **5.74** | 5.82 | **5.71** | 50 |
| | AMR | 1.32 | 1.57 | 107 | 1.85 | 5.75 | 5.75 | 5.75 | **5.75** | **0.65** |
| | AC | **0.199** | **0.0612** | **95** | **1.45** | 6.66 | 6.71 | 7.138 | 6.54 | 1.69 |
| | Baseline | 1.763 | 2.46 | 18372 | 2.51 | 5.672 | 5.923 | 5.825 | 5.752 | 100 |
| Communities | MR | 26.74 | 28863 | - | - | 0.023 | 0.026 | - | - | 75 |
| | AMR | **4.36** | **53.15** | - | - | 0.020 | **0.024** | - | - | **0.20** |
| | AC | - | - | - | - | - | - | - | - | - |
| | Baseline | 26.72 | 33475 | - | - | **0.019** | 0.024 | - | - | 100 |

integrates data repair with stochastic gradient descent: in each iteration it samples a batch, returns it to the user for repair, and then updates model parameters with the repaired samples. Although *AC* reduces repair cost by prioritizing influential samples, it is unclear whether the resulting repaired data yield an accurate model, since not all samples are ever selected for gradient updates. In these experiments, we use datasets with injected missingness with ground truth to simulate manual repairs.

Table 3 reports SVM classification results for minimal repair (*MR*), almost minimal repair (*AMR*), and *AC*. The results show that *MR* and *AMR* consistently outperform *AC* in accuracy across all datasets and missingness levels. Notably, *AMR* achieves higher accuracy than *AC* while repairing substantially fewer samples. *MR* has the highest accuracy overall, although it selects more samples.

Table 5 compares regression outcomes for *MR*, *AMR*, and *AC*. Unlike *AC* and *AMR*, which repair entire samples, *MR* imputes individual missing features (see Section 4). Consistent with the classification findings, *MR* and *AMR* again outperform *AC* in terms of mean squared error (MSE) across all datasets and missingness ratios. *MR* achieves the lowest MSE overall, reflecting its more comprehensive repair strategy aimed at closely approximating the optimal model. The results for SVM and linear regression on these datasets with types of missingness show a similar trend and are in the appendix.

Next, we evaluate the time and effort saved by our methods using model-based imputations for repair. Because the imputation cost increases with the number of missing items, (almost) minimal repair can cut both inference time and user effort for inspecting or verifying imputed values.We use four imputation models that span the major methodological families. *KNN* Mattei & Frellsen (2019) represents a classical distance-based approach that predicts missing values from nearby observed samples. *MICE* Buuren & Groothuis-Oudshoorn (2011) provides a statistical baseline based on multivariate regression and remains widely used in practical data-analysis workflows. *MissForest* Stekhoven & Bühlmann (2011) is a non-parametric machine-learning method that leverages random forests to capture nonlinear dependencies. Finally, *TCSDI* Zheng & Charoenphakdee (2022). serves as our modern deep generative baseline; as a diffusion-based imputer, it has been shown to outperform earlier deep-learning methods such as GAIN and VAE-based models. Together, these four methods cover the statistical, traditional ML and deep generative paradigms, providing a representative spectrum of imputation strategies. Across all models, we evaluate four imputation regimes, full imputation, minimal-repair imputation, (almost) minimal-repair imputation, and ActiveClean-selected sample imputation and report accuracy, running time, and the number of imputed items.

As shown in Table B, our methods are generally faster and deliver higher accuracies than *AC* and full imputations over datasets with original missingness and different imputation models for SVM. *TCSDI* achieves a higher accuracy than other imputation models, but with longer inference times than other methods. This underscores the practical value of *MR* and *AMR*, which substantially reduce inference overhead by limiting imputations, especially when paired with *TCSDI*. The results for SVM over datasets with injected missingness and different model-based imputation methods show a similar trend and are in the appendix.

We also assess linear regression with model-based imputations (Table E). Some imputation methods run out of main memory over some datasets, e.g., *MF* on the *Communities* dataset, as they scale poorly on datasets with too many features/samples. We have omitted their results in their corresponding tables (shown as -). Here, *MR* and *AMR* generally deliver faster inference than full imputation while maintaining comparable accuracy, despite substantially fewer imputations. In contrast, *AC* encountered computational challenges in datasets with high missingness ratios, e.g., Communities, where minimal cleaning occasionally leaves zero training samples, causing failures in partial fitting. *MR* and *AMR* avoid such failures, demonstrating robustness at substantial missingness ratios. *AC* also generally delivers higher MSE (lower fit) than *MR* and *AMR*. The results for linear regression over datasets with injected missingness show a similar trend and are in the appendix.

Finally, while some of our theoretical results assume conditions such as zero-mean Gaussian noise or M-Lipschitz continuity of loss functions, we observe that these assumptions are not required in practice. The datasets in our empirical evaluation do not satisfy these conditions, and SVM models do not satisfy M-Lipschitz continuity; nonetheless, *MR* and *AMR* consistently deliver accurate results.

## 7 AMR for Neural Networks

Computing AMR for Deep Neural Networks (DNNs) is challenging as minimizing the suboptimality gap requires global minima for every repair. Recall that an ACM $\mathbf{w}^{\approx}$ satisfies:

$$\sup_{\mathbf{X}^r \in \mathbf{X}^R} \left( L(\mathbf{w}^{\approx}, \mathbf{X}^r, \mathbf{y}) - \min_{\mathbf{w} \in \mathcal{W}} L(\mathbf{w}, \mathbf{X}^r, \mathbf{y}) \right) \le e. \tag{5}$$

Since flexible DNNs can memorize random data with near-zero loss Zhang et al. (2016); Cooper (2018), we assume $\min_{\mathbf{w} \in \mathcal{W}} L(\mathbf{w}, \mathbf{X}^r, \mathbf{y}) \approx 0$. Thus, checking for an ACM simplifies to ensuring the worst-case loss is bounded.

To make this feasible, we check the condition sample-wise. We verify if the average difference between maximal and minimal repair losses satisfies the threshold $e$:

$$\frac{1}{n} \sum_{\mathbf{x}_i \in MVX} \left( \max_{\mathbf{x}_i^r \in \mathbf{x}_i^R} L(f(\mathbf{x}_i^r, \mathbf{w}^\approx), y_i) - \min_{\mathbf{x}_i^r \in \mathbf{x}_i^R} L(f(\mathbf{x}_i^r, \mathbf{w}^\approx), y_i) \right) \leq e \qquad (6)$$

For networks with monotonic activations (e.g., ReLU), extrema occur at missing value boundaries (edge repairs), significantly reducing cost.

When the ACM condition is violated, AMR is the smallest subset restoring it. Intuitively, we impute samples contributing most to the loss difference in Eq. 6. Since this ranking is model-dependent, Algorithm 2 collaboratively learns the DNN and identifies AMR via SGD.

---

**Algorithm 2** Approximating AMR for Neural Networks

---

$S_{AMR} \leftarrow [\ \ ]$
$\mathbf{X}^{curr} \leftarrow$ random repair of $\mathbf{X}$; $\mathbf{w}^{curr} \leftarrow$ random initialization
$maxL, minL, samplesTopK \leftarrow checkACM(\mathbf{w}^{curr}, \mathbf{X}^{curr})$
**while** $maxL - minL > e$ **do**
  **if** $maxL > e$ **AND** $minL \leq e$ **then**
    $\mathbf{X}^{curr} \leftarrow impute(\mathbf{X}^{curr}, samplesTopK)$
    $S_{AMR} \leftarrow S_{AMR}.add(samplesTopK)$
  **end if**
  $\mathbf{w}^{curr} \leftarrow SGDEpoch(\mathbf{w}^{curr}, \mathbf{X}^{curr})$
  Update $maxL, minL, samplesTopK$
**end while**
**return** $S_{AMR}$

---

The algorithm iteratively identifies $k$ samples to restore the ACM condition. Crucially, we only impute when the model is "approximately optimal" for some repairs ($minL \leq e < maxL$). If both losses exceed $e$, we prioritize training ($SGDEpoch$) to ensure selection reflects true sample importance. $S_{AMR}$ is the union of imputed samples.

## 8 RELATED WORK

Researchers have proposed *stochastic optimization* to find a model by optimizing the expected loss function over the probability distributions of missing data items in training samples Ganti & Willett (2015). Similarly, *robust optimization* aims to minimize the loss function of a model for the imputation that brings the highest training loss given certain distributions of missing values Aghasi et al. (2022). However, the distributions of missing data items are not often available. Thus, users may spend significant time and effort discovering or training these distributions, which may require the user to find the causes of missingness in the data and dependencies between the features. Additionally, for a given type of model, users must solve various and possibly challenging optimization problems for many possible (combinations of) distributions of missing values. More importantly, these methods reflect the uncertainty in the training data caused by missing values in the trained model instead of repairing the data to reduce its uncertainty. Hence, they deliver inaccurate models on the dataset with many missing values. More discussion about related work is available in the appendix.

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

## LIMITATIONS

While our work demonstrates both theoretical and practical advantages in learning over incomplete data, we acknowledge two limitations:

**Model Class and Convexity Assumptions.** Our proposed minimal repair (MR) algorithms are developed for support vector machines (SVM) and linear regression, while the almost minimal repair (AMR) framework is applicable to a broader class of statistical machine learning models. However, for AMR, we currently provide provable error bounds and efficient approximations only for models with convex loss functions. This stems from our reliance on Step 1 (ST1) in Section 5.1, where we solve a convex optimization problem to find an approximately optimal model $w_k^{\widetilde{\approx}}$. Extending AMR to models with non-convex loss functions remains an open challenge due to the difficulty of verifying approximate optimality in such settings. Importantly, this limitation reflects the well-known hardness of non-convex optimization itself—since one cannot generally find globally optimal models for non-convex losses, it is also difficult to guarantee that a repaired model is close to a global optimum.

**Trade-off Between Computation and Imputation Time.** As seen in our experiments, the time required to compute MR or AMR can exceed or roughly match the time needed to fully impute the dataset when simple imputation methods (e.g., mean or KNN) are used. This suggests that MR and AMR may not be the preferred choice in scenarios where users already opt for inexpensive imputation strategies. However, for more complex, resource-intensive and often accurate imputation methods—such as diffusion-based models Zheng & Charoenphakdee (2022)—we observe substantial time savings by using MR or AMR to reduce the number of imputations. In practice, users may choose to apply MR or AMR when planning to use high-cost imputation models, and directly pursue full imputation when using simpler methods.

## ROBUSTNESS TO IMPUTATION ERROR

While our study focuses on the impact of different imputation strategies on downstream model performance, an important complementary direction is the development of learning methods that remain robust to potential imputation errors. This robustness-focused perspective is largely orthogonal to our goal of understanding how different subsets of data influence imputation quality, and it would require a substantial separate investigation beyond the scope of this work. Approaches based on robust optimization, for example, offer a principled way to train models that account for uncertainty in the imputed values and may help mitigate the effects of imputation variability. Exploring such robustness-oriented techniques represents a promising avenue for future work.

## BROADER IMPACTS

Our work has potential positive and negative societal impacts, which we outline below.

**Positive societal impacts.** Our methods can substantially reduce the time and effort needed for data preparation, a phase that often consumes up to 80% of a data scientist's time Neutatz et al. (2021). By identifying only the essential missing values to repair, our approach streamlines the ML pipeline, lowers costs, and makes ML more accessible for everyone—especially in resource-constrained settings or domains where full imputation is infeasible.

**Negative societal impacts.** In high-stakes domains, e.g., healthcare, criminal justice, setting a suboptimal error threshold in AMR (either intentionally or unintentionally) may lead to missed repairs of critical data, resulting in biased or unsafe models. Additionally, the selective repair approach may cause developers to overlook the importance of understanding missingness mechanisms or domain context. These risks can be mitigated by involving domain experts and validating models before deployment.

## EXPERIMENTAL SETTING

### DATASETS

We evaluate our methods on two types of datasets: those with synthetic missingness and those with real-world missingness. For each dataset, we simulate three levels of missingness: 0.2, 0.4, and 0.6, corresponding to 20%, 40%, and 60% incomplete samples, respectively. These datasets are further divided based on the downstream task: linear regression (LR) and support vector machine classification (SVM).

All datasets are obtained from publicly available repositories. For synthetic missingness, we start with complete datasets and introduce missing values in a controlled manner. For real missingness, we use datasets that naturally contain incomplete entries. This separation allows us to analyze the behavior of our repair methods under both idealized and realistic data corruption scenarios.

### DETECTION OF TYPES OF MISSINGNESS IN DATA

We use the Missing Value PC (MVPC) algorithm Tu et al. (2019), a framework designed for causal discovery in datasets with missing data. It is an extension of the PC algorithm, which is a constraint-

based method for causal discovery. Given an incomplete dataset, we introduce a missingness indicator $R_A$ for each incomplete feature $A$. We run the Missing Value PC (MVPC) algorithm on the dataset, after including the indicators, and we inspect the dependencies of $R_A$.

## HARDWARE

We conducted experiments on two hardware platforms. Most experiments ran on an x86_64 machine with 30 Intel(R) Xeon(R) E5-2670 v3 CPU cores (2.30GHz), hosted in a VMware virtualized environment with two NUMA nodes and 30MB L3 cache. However, this system lacked sufficient power for diffusion-based imputation models. For those experiments (TCSDI), we used an Nvidia DGX-2 system with one Nvidia Tesla V100 GPU (32GB VRAM) and 20 CPU cores from 2.70GHz Intel Xeon Platinum 8168 processors with 33MB L3 cache.

## USING SGD

We have run each experiment that involves Stochastic Gradient Descent (SGD) (SVM) three times with different seeds and report the average.

## ADDITIONAL EXPERIMENTAL RESULTS

Tables B, C, D and E report the accuracy and running times of full imputation (baseline), *MR*, *AMR*, and *AC* across all imputation methods. For SVM, we evaluate all three injected missingness types (MCAR, MAR, MNAR), while for linear regression we include the MCAR setting due to feasibility constraints. Across these experiments, the results follow the same trend described in Section 7.

## ADDITIONAL MEMORY-USAGE RESULTS

We report the reduction in computation time for imputing MR and AMR subsets relative to full-data imputation in Table B of the submission. For memory usage, only the Malware–MAR results are included due to space constraints (shown in the table F, G, I). These partial results already illustrate a consistent trend: KNN exhibits substantially lower peak RAM consumption under MR compared to full imputation, and MissForest also shows reduced peak memory across the malware20, malware40, and malware60 configurations. MICE is omitted because it was infeasible to run on this dataset within the available memory budget.

Overall, the observed pattern suggests that MR generally lowers memory usage, particularly for distance-based and non-parametric methods whose resource requirements scale with the number of samples. Future extensions may include a broader memory-usage comparison across additional datasets and missingness settings.

## MINIMAL REPAIR FOR LINEAR REGRESSION

### ALGORITHM FOR FINDING MINIMAL REPAIR

Orthogonal Matching Pursuit (OMP) provides an efficient approximation for solving the sparse linear regression problem Wang et al. (2012). Essentially, this greedy algorithm begins with an empty solution set and initializes the regression residual to the label vector. In each iteration, the algorithm selects the feature most relevant to the current residual (i.e., having the largest dot product), adds it to the solution set, retains a linear regression model, and updates the residual accordingly. The program stops when the regression residue is sufficiently small. Therefore, OMP will return a subset of features (the solution set) that are sufficient to achieve an optimal linear regression model.

In this paper, we propose a variant of OMP, as outlined in Algorithm A, to find minimal repair for linear regression. Our algorithm has two major differences compared to the conventional OMP. Firstly, we include all complete features in the regression at the initialization, ensuring that we minimize the number of non-zero coefficients only among incomplete features. Secondly, we define our stopping condition by the maximum relevance (cosine similarity) between the feature and the label being

Table B: Accuracy/time for SVM on data with injected MCAR using model-based imputations

| Data Set | Method | Time(s) | | | | Accuracy(%) | | | | % Samples Imputed |
|---|---|---|---|---|---|---|---|---|---|---|
| | | KNN | MICE | TCSDI | MF | KNN | MICE | TCSDI | MF | |
| Malware 20 | MR | 8.98 | - | 24390.8 | 4534.6 | 95.6 | - | 96.49 | 95.8 | 18.68 |
| | AMR | 19.13 | - | 953.1 | 26.23 | 95.42 | - | 95.78 | 95.31 | 0.73 |
| | AC | 1.17 | - | 12668 | 3274 | 91.20 | - | 92.30 | 91.60 | 9.09 |
| | Baseline | 13.9 | - | 130269 | 6645.5 | 96.52 | - | 96.74 | 96.70 | 100 |
| Malware 40 | MR | 14.24 | - | 54702.4 | 3924.4 | 96.16 | - | 96.23 | 96.17 | 21.1 |
| | AMR | 9.99 | - | 1057.3 | 25.7 | 95.89 | - | 96.23 | 91.88 | 0.37 |
| | AC | 0.84 | - | 11085 | 3078.5 | 88.97 | - | 89.73 | 89.57 | 5.32 |
| | Baseline | 27.66 | - | 260537 | 6347.8 | 93.83 | - | 96.74 | 95.75 | 100 |
| Malware 60 | MR | 18.26 | - | 64959.2 | 3546.5 | 96.16 | - | 97.87 | 96.41 | 16.65 |
| | AMR | 11.28 | - | 1107.2 | 23.91 | 95.12 | - | 96.01 | 94.37 | 0.24 |
| | AC | 0.84 | - | 11085 | 2534.7 | 89.00 | - | 88.82 | 89.32 | 3.28 |
| | Baseline | 41.21 | - | 390806 | 6256.4 | 96.16 | - | 97.87 | 96.70 | 100 |
| Tuadromd 20 | MR | 1.39 | 100.8 | 287 | 41.4 | 98.6 | 98.73 | 98.43 | 98.67 | 11.9 |
| | AMR | 2.81 | 17.28 | 43.74 | 6.01 | 96.13 | 96.27 | 96.89 | 96.13 | 2.02 |
| | AC | 1.04 | 99.86 | 145 | 15.3 | 97.58 | 97.63 | 97.63 | 97.63 | 4.06 |
| | Baseline | 2.37 | 102.11 | 1987 | 45.60 | 98.77 | 98.77 | 98.66 | 98.70 | 100 |
| Tuadromd 40 | MR | 2.55 | 96.26 | 466.4 | 36.69 | 98.5 | 98.4 | 98.54 | 98.5 | 11.1 |
| | AMR | 2.94 | 18.15 | 83.64 | 3.87 | 96.15 | 95.78 | 96.56 | 96.15 | 1.01 |
| | AC | 0.86 | 99.8 | 169 | 13.4 | 96.98 | 97.12 | 97.45 | 97.45 | 3.3 |
| | Baseline | 4.69 | 100.14 | 3882 | 43.94 | 97.3 | 97.6 | 98.66 | 98.84 | 100 |
| Tuadromd 60 | MR | 3.62 | 79.66 | 692.6 | 33.41 | 98.3 | 98.36 | 98.38 | 98.3 | 11.8 |
| | AMR | 6.52 | 9.13 | 137.5 | 25.13 | 95.45 | 95.25 | 96.13 | 95.98 | 0.67 |
| | AC | 0.679 | 77.08 | 170 | 10.5 | 96.96 | 96.88 | 97.3 | 96.90 | 1.87 |
| | Baseline | 6.21 | 100.6 | 6476 | 43.96 | 97.6 | 97.3 | 98.66 | 98.66 | 100 |
| Credit Default 20 | MR | 8.37 | 2.93 | 2121 | 98.24 | 80.0 | 78.1 | 79.6 | 81.07 | 30 |
| | AMR | 11.05 | 15.48 | 21.37 | 63.25 | 78.10 | 78.14 | 78.10 | 78.87 | 0.08 |
| | AC | 14.49 | 15.43 | 94 | 64.8 | 78.3 | 78.16 | 78.2 | 78.25 | 0.125 |
| | Baseline | 23.20 | 5.14 | 7071 | 108.3 | 78.1 | 80.1 | 80.3 | 78.40 | 100 |
| Credit Default 40 | MR | 11.77 | 2.64 | 4242.6 | 76.45 | 80.31 | 80.1 | 80.4 | 80.85 | 30 |
| | AMR | 12.37 | 16.75 | 29.57 | 51.27 | 78.14 | 78.12 | 78.12 | 78.22 | 0.04 |
| | AC | 18.07 | 18.78 | 96 | 45.35 | 79.76 | 79.1 | 80.01 | 79.87 | 0.19 |
| | Baseline | 38.56 | 5.05 | 14263 | 98.32 | 79.6 | 78.1 | 78.08 | 78.14 | 100 |
| Credit Default 60 | MR | 13.56 | 3.5 | 6357 | 58.4 | 79.72 | 79.81 | 79.75 | 80.32 | 30 |
| | AMR | 14.12 | 15.79 | 32.15 | 29.35 | 78.14 | 78.12 | 78.12 | 78.09 | 0.03 |
| | AC | 20.87 | 21.31 | 94 | 35.7 | 79.4 | 79.3 | 79.7 | 79.4 | 0.21 |
| | Baseline | 48.04 | 3.902 | 21124 | 89.17 | 79.1 | 79.6 | 80.1 | 78.12 | 100 |

Table C: Accuracy/time for SVM on data with injected MAR using model-based imputations

| Data Set | Method | Time(s) | | | | Accuracy(%) | | | | Impute (%) |
|---|---|---|---|---|---|---|---|---|---|---|
| | | KNN | MICE | TCSDI | MF | KNN | MICE | TCSDI | MF | |
| Malware 20 | MR | 11.16 ± 0.89 | - | 27135 ± 2453.1 | 4726.8 ± 466.2 | 96.38 ± 0.25 | - | 96.45 ± 0.13 | 96.29 ± 0.34 | 21 |
| | AMR | 20.37 ±1.54 | - | 989.4 ±11.74 | 31.54 ±2.03 | 95.17 ±0.13 | - | 95.87 ±0.02 | 95.03 ±0.75 | 0.73 ±0 |
| | AC | 2.291 ± 0.049 | - | 5295.3 ± 113.3 | 4177 ± 6.23 | 93.39 ± 1.54 | - | 94.85 ± 0.20 | 93.97 ± 0.96 | 3.34 ± 0.15 |
| | Baseline | 17.71 | - | - | 6830.2 | 96.30 | - | - | 96.59 | 100 |
| Malware 40 | MR | 19.11 ± 1.00 | - | 65610 ± 3524.5 | 5422.8 ± 835.9 | 96.16 ± 0.24 | - | 96.13 ± 0.06 | 96.10 ± 0.17 | 25.23 |
| | AMR | 19.98 ±1.24 | - | 1043.9 ±53.2 | 35.37 ±1.3 | 96.03 ±0.02 | - | 95.89 ±0.04 | 95.36 ±0.13 | 0.37 ±0 |
| | AC | 2.178 ± 0.106 | - | 10534 ± 1322.7 | 5168.1 ± 90.4 | 94.55 ± 0.95 | - | 93.89 ± 0.48 | 92.14 ± 0.83 | 3.29 ± 0.22 |
| | Baseline | 36.10 | - | - | 6460.7 | 96.15 | - | - | 95.90 | 100 |
| Malware 60 | MR | 23.44 ± 0.88 | - | 76545 ± 5246.8 | 4224.5 ± 549.4 | 95.67 ± 0.61 | - | 95.73 ± 0.10 | 95.65 ± 1.09 | 19.69 |
| | AMR | 17.37 ±0.87 | - | 1276.3 ±57.2 | 24.1 ±0.8 | 95.71 ±0.52 | - | 95.46 ±0.21 | 94.01 ±0.12 | 0.24 ±0 |
| | AC | 1.807 ± 0.036 | - | 15259 ± 1631.2 | 2968.5 ± 25.04 | 93.44 ± 1.35 | - | 93.01 ± 0.67 | 91.78 ± 2.45 | 3.24 ± 0.23 |
| | Baseline | 48.33 | - | - | 5794.3 | 95.85 | - | - | 96.18 | 100 |
| Tuadromd 20 | MR | 1.21 ± 0.003 | 98.65 ± 0.85 | 305.8 ± 5.13 | 39.74 ± 0.62 | 98.68 ± 0.18 | 98.63 ± 0.15 | 98.70 ± 0.11 | 98.67 ± 0.20 | 9.84 |
| | AMR | 2.98 ±0.13 | 20.01 ±1.26 | 51.27 ±2.01 | 6.99 ±0.43 | 95.24 ±0.23 | 96.03 ±0.24 | 96.18 ±0.21 | 95.78 ±0.14 | 2.02 ±0 |
| | AC | 0.907 ± 0.052 | 100.85 ± 3.51 | 133.6 ± 6.84 | 39.10 ± 0.56 | 98.76 ± 0.10 | 98.76 ± 0.16 | 98.89 ± 0.07 | 98.66 ± 0.07 | 3.87 ± 0.76 |
| | Baseline | 1.72 | 101.1 | 1673.4 | 44.55 | 98.80 | 98.73 | 98.81 | 98.81 | 100 |
| Tuadromd 40 | MR | 2.26 ± 0.01 | 95.7 ± 0.55 | 492.5 ± 10.4 | 37.2 ± 0.23 | 98.73 ± 0.11 | 98.67 ± 0.10 | 98.75 ± 0.09 | 98.78 ± 0.06 | 10.19 |
| | AMR | 3.23 ±0.09 | 20.17 ±1.03 | 46.24 ±2.12 | 6.09 ±0.53 | 96.35 ±0.27 | 96.07 ±0.45 | 96.84 ±0.25 | 96.87 ±0.19 | 1.01 ±0 |
| | AC | 0.686 ± 0.013 | 93.65 ± 1.57 | 138.33 ± 4.92 | 34.52 ± 0.22 | 98.73 ± 0.11 | 98.70 ± 0.07 | 98.76 ± 0.07 | 98.78 ± 0.11 | 3.01 ± 0.66 |
| | Baseline | 3.17 | 101.5 | 3451 | 44.57 | 98.70 | 98.81 | 98.80 | 98.70 | 100 |
| Tuadromd 60 | MR | 3.27 ± 0.03 | 82.4 ± 5.56 | 671 ± 35.8 | 35.13 ± 0.11 | 98.86 ± 0.12 | 98.71 ± 0.02 | 98.83 ± 0.11 | 98.73 ± 0.12 | 9.62 |
| | AMR | 6.78 ±0.12 | 12.53 ±1.17 | 124.3 ±4.54 | 28.71 ±1.31 | 96.99 ±0.17 | 97.13 ±0.26 | ±97.02 ±0.35 | 97.54 ±0.43 | 0.67 ±0 |
| | AC | 0.556 ± 0.010 | 74.63 ± 1.10 | 165.7 ± 8.49 | 31.68 ± 0.24 | 98.64 ± 0.19 | 98.60 ± 0.09 | 98.49 ± 0.13 | 98.61 ± 0.02 | 2.32 ± 0.12 |
| | Baseline | 4.39 | 116.08 | 5036 | 44.17 | 98.82 | 98.71 | 98.82 | 98.73 | 100 |
| Credit Default 20 | MR | 5.54 ± 1.18 | 3.71 ± 0.68 | 2137.5 ± 41.5 | 100.6 ± 0.73 | 81.05 ± 0.03 | 81.08 ± 0.02 | 81.12 ± 0.04 | 81.12 ± 0.02 | 30 |
| | AMR | 10.75 ± 0.37 | 17.03 ± 0.88 | 20.13 ± 0.95 | 71.35 ± 0.17 | 79.43 ± 0.17 | 79.32 ± 0.08 | 79.478 ± 0.15 | 79.02 ± 0.09 | 0.08 ± 0 |
| | AC | 12.57 ± 1.06 | 13.58 ± 1.03 | 72.44 ± 5.01 | 104.12 ± 1.16 | 81.02 ± 0.00 | 81.02 ± 0.00 | 81.02 ± 0.00 | 81.02 ± 0.00 | 0.56 ± 0.30 |
| | Baseline | 11.45 | 6.14 | 7125 | 112.88 | 81.02 | 81.02 | 81.02 | 81.02 | 100 |
| Credit Default 40 | MR | 9.55 ± 1.51 | 5.54 ± 0.74 | 4275.1 ± 34.7 | 79.87 ± 1.30 | 56.68 ± 0.00 | 56.68 ± 0.00 | 56.68 ± 0.30 | 56.70 ± 0.00 | 30 |
| | AMR | 12.98 ± 0.57 | 19.03 ± 0.16 | 35.13 ± 0.77 | 41.02 ± 2.21 | 79.13 ± 0.06 | 79.99 ± 0.05 | 80.01 ± 0.11 | 79.76 ± 0.08 | 0.04 |
| | AC | 13.50 ± 1.56 | 14.17 ± 1.95 | 83.56 ± 4.67 | 80.14 ± 1.45 | 78.16 ± 0.05 | 78.17 ± 0.05 | 78.21 ± 0.07 | 78.17 ± 0.04 | 0.52 ± 0.20 |
| | Baseline | 19.93 | 6.15 | 14291 | 106.22 | 81.02 | 81.02 | 81.02 | 81.02 | 100 |
| Credit Default 60 | MR | 7.14 ± 0.26 | 4.12 ± 0.72 | 6411 ± 45.4 | 66.12 ± 2.43 | 60.59 ± 0.04 | 60.65 ± 0.05 | 60.64 ± 0.09 | 60.65 ± 0.03 | 30 |
| | AMR | 15.74 ± 0.27 | 19.13 ± 1.08 | 41.98 ± 1.56 | 32.17 ± 1.10 | 78.32 ± 0.04 | 78.88 ± 0.07 | 79.27 ± 0.02 | 79.03 ± 0.03 | 0.03 |
| | AC | 7.70 ± 2.37 | 8.23 ± 1.97 | 84.85 ± 4.51 | 52.06 ± 2.26 | 81.04 ± 0.01 | 81.03 ± 0.01 | 81.04 ± 0.02 | 81.03 ± 0.01 | 0.30 ± 0.13 |
| | Baseline | 26.07 | 7.91 | 21307 | 103.05 | 81.02 | 81.02 | 81.02 | 81.02 | 100 |

smaller than or equal to a user-defined threshold, instead of relying on a near-zero regression residue. This approach enables our algorithm to work with general datasets without requiring the assumption of an underdetermined linear system, which is typically necessary in conventional OMP.

Table D: Accuracy/time for SVM on data with injected MNAR using model-based imputations

| Data Set | Method | Time(s) | | | | Accuracy(%) | | | | Impute (%) |
|---|---|---|---|---|---|---|---|---|---|---|
| | | KNN | MICE | TCSDI | MF | KNN | MICE | TCSDI | MF | |
| Malware 20 | MR | 15.49±0.70 | - | 43605±2020.5 | 5206.1±812.6 | 96.07±0.44 | - | 96.20±0.34 | 96.10±0.6 | 18.49±0.85 |
| | AMR | 19.14 ± 0.75 | - | 879.2 ± 23.5 | 29.10 ± 1.89 | 95.14 ± 0.04 | - | 95.76 ± 0.02 | 95.57 ± 0.03 | 0.73 ± 0 |
| | AC | 2.211±0.12 | - | 6619±1161.3 | 5292 ± 110.6 | 92.37±0.746 | - | 92.74 ± 0.45 | 92.68±1.68 | 3.494±0.567 |
| | Baseline | 28.96 | - | - | 5959.7 | 96.55 | - | - | 96.52 | 100 |
| Malware 40 | MR | 25.32±1.30 | - | 95044±5955.3 | 3622.7±167.1 | 95.16±0.76 | - | 95.13±0.4 | 95.02±0.1 | 22.83±1.4 |
| | AMR | 21.37 ± 1.12 | - | 1198.3 ± 45.7 | 29.17 ± 4.23 | 95.11 ± 0.05 | - | 95.18 ± 0.05 | 94.98 ± 0.05 | 0.37 ± 0 |
| | AC | 1.525±0.096 | - | 10534±1322.7 | 1659.4±3.94 | 93.99±1.747 | - | 93.48±0.48 | 93.63±0.63 | 3.245±0.121 |
| | Baseline | 50.34 | - | - | 7124.4 | 96.52 | - | - | 96.33 | 100 |
| Malware 60 | MR | 27±1.94 | - | 103279±8018.5 | 1598.4±243.7 | 95.23±0.40 | - | 95.20±0.21 | 95.13±0.17 | 19.75±2.66 |
| | AMR | 21.37 ± 1.24 | - | 1492.3 ± 98.5 | 22.13 ± 1.07 | 95.34 ± 0.05 | - | 94.58 ± 0.06 | 95.02 ± 0.08 | 0.24 ± 0 |
| | AC | 0.946±0.0617 | - | 15259±1631.2 | 1128.5±8.95 | 94.27±0.341 | - | 94.01±0.67 | 94.04±0.30 | 3.363±0.174 |
| | Baseline | 63.30 | - | - | 6061.5 | 96.18 | - | - | 96.08 | 100 |
| Tuadromd 20 | MR | 1.98±0.06 | 96.31±1.12 | 314.6±13.6 | 37.55±0.24 | 98.74±0.06 | 98.69±0.13 | 98.80±0.12 | 98.71±0.1 | 10.11±1.55 |
| | AMR | 2.17 ± 0.55 | 18.96 ± 2.13 | 47.22 ± 1.35 | 6.13 ± 0.54 | 97.84 ± 0.03 | 97.94 ± 0.05 | 98.28 ± 0.03 | 98.22 ± 0.06 | 2.02 ± 0 |
| | AC | 0.785±0.057 | 94.39±2.04 | 133.6±6.84 | 35.82±0.059 | 98.64±0.033 | 98.65±0.092 | 98.69±0.07 | 98.62±0.03 | 2.99±0.755 |
| | Baseline | 2.55 | 100.5 | 3402.6 | 46.34 | 98.80 | 98.74 | 98.82 | 98.88 | 100 |
| Tuadromd 40 | MR | 3.37±0.01 | 78.93±1.45 | 612.3±15.6 | 34.62±0.24 | 98.56±0.05 | 98.62±0.19 | 98.75±0.1 | 98.36±0.02 | 9.95±0.22 |
| | AMR | 2.95 ± 0.87 | 23.49 ± 3.15 | 50.27 ± 7.16 | 7.02 ± 0.99 | 97.94 ± 0.13 | 97.76 ± 0.10 | 98.08 ± 0.12 | 97.84 ± 0.09 | 1.01 ± 0 |
| | AC | 0.515±0.004 | 94.39±2.00 | 138.3±4.92 | 31.40±0.21 | 98.21±0.13 | 98.11±0.25 | 98.26±0.1 | 98.13±0.097 | 2.163±0.182 |
| | Baseline | 4.39 | 103.71 | 6098.7 | 44.13 | 98.70 | 98.87 | 98.87 | 98.80 | 100 |
| Tuadromd 60 | MR | 4.31±0.01 | 72.36±1.19 | 751.2±21.4 | 33.77±0.2 | 98.1±0.16 | 98.0±0.16 | 98.14±0.13 | 98.03±0.12 | 9.39±0.27 |
| | AMR | 6.99 ± 0.83 | 17.13 ± 3.57 | 155.3 ± 20.7 | 26.39 ± 4.13 | 97.78 ± 0.10 | 98.12 ± 0.12 | 98.27 ± 0.11 | 98.05 ± 0.09 | 0.67 ± 0 |
| | AC | 0.248±0.053 | 65.56±2.93 | 165.7±8.49 | 29.63±0.24 | 97.89±0.114 | 97.94±0.102 | 97.94±0.13 | 97.58±0.45 | 2.138±0.125 |
| | Baseline | 5.73 | 101.39 | 6090.7 | 43.86 | 98.84 | 98.59 | 98.82 | 98.82 | 100 |
| Credit Default 20 | MR | 6.13±0.33 | 3.05±0.27 | 3827±57.1 | 87.80±0.08 | 80.91±0.07 | 80.85±0.004 | 80.88±0.005 | 80.85±0.004 | 29.71 |
| | AMR | 11.43 ± 0.75 | 20.01 ± 1.85 | 25.37 ± 1.29 | 59.95 ± 4.27 | 80.28 ± 0.12 | 80.75 ± 0.09 | 81.32 ± 0.08 | 80.45 ± 0.11 | 0.08 ± 0 |
| | AC | 14.19±1.15 | 15.37±1.15 | 13.78±1.15 | 118.3±13.27 | 79.90±0.923 | 79.74±0.834 | 79.87 ± 0.768 | 79.71±0.88 | 0.486±0.246 |
| | Baseline | 17.67 | 6.65 | 12820.5 | 113.13 | 81.02 | 81.02 | 81.02 | 81.02 | 100 |
| Credit Default 40 | MR | 7.43±0.61 | 3.89±0.76 | 6850±76.4 | 60.97±0.15 | 60.37±0.03 | 58.90±0.18 | 60.64±0.14 | 60.47±0.04 | 29.98 |
| | AMR | 13.07 ± 0.78 | 21.54 ± 1.76 | 41.29 ± 6.72 | 45.24 ± 2.92 | 79.52 ± 0.10 | 80.03 ± 0.14 | 80.05 ± 0.07 | 80.18 ± 0.12 | 0.04 ± 0 |
| | AC | 9.94±0.243 | 10.725±0.243 | 9.68±0.243 | 71.85±1.93 | 80.96±0.056 | 80.92±0.13 | 80.98 ± 0.005 | 81.01±0.004 | 0.327±0.095 |
| | Baseline | 29.59 | 6.25 | 22838 | 105.59 | 81.02 | 81.02 | 81.02 | 81.02 | 100 |
| Credit Default 60 | MR | 14.07±1.41 | 5.8±0.32 | 8992±61.4 | 47.25±0.03 | 58.95±0.20 | 58.5±0.1 | 60.35±0.19 | 60.43±0.25 | 29.89 |
| | AMR | 16.83 ± 0.72 | 25.19 ± 3.61 | 49.17 ± 2.43 | 44.54 ± 2.47 | 79.73 ± 0.13 | 80.03 ± 0.07 | 81.03 ± 0.11 | 80.89 ± 0.13 | 0.03 ± 0 |
| | AC | 4.45±0.49 | 4.79±0.49 | 4.38±0.49 | 34.04±0.69 | 80.56±0.597 | 80.52±0.594 | 80.65 ± 0.34 | 80.69±0.39 | 0.213±0.021 |
| | Baseline | 35.90 | 6.77 | 29978.4 | 104.16 | 81.02 | 81.02 | 81.02 | 81.02 | 100 |

Table E: Accuracy/time for Linear Regression on data with injected MCAR using model-based imputations

| Data Set | Method | Time(s) | | | | MSE | | | | % Impute Samples |
|---|---|---|---|---|---|---|---|---|---|---|
| | | KNN | MICE | TCSDI | MF | KNN | MICE | TCSDI | MF | |
| Superconductivity 20 | MR | 2.369 | 33.16 | 29165 | 851.1 | 0.0089 | 0.00897 | 0.00904 | 0.0091 | 70 |
| | AMR | 25.13 | 30.70 | 39.95 | 29.78 | 0.0247 | 0.0247 | 0.0247 | 0.0247 | 0.09 |
| | AC | 0.145 | 15.66 | 96.4 | 515.92 | 0.00934 | 0.0092 | 0.0092 | 0.00928 | 0.05 |
| | Baseline | 0.0692 | 34.17 | 30515 | 794.8 | 0.0088 | 0.008939 | 0.009013 | 0.0091 | 100 |
| Superconductivity 40 | MR | 2.599 | 33.73 | 29735 | 842.4 | 0.0089 | 0.00924 | 0.00914 | 0.0092 | 75.00 |
| | AMR | 50.77 | 56.64 | 85.61 | 60.24 | 0.0314 | 0.0314 | 0.0314 | 0.0314 | 0.05 |
| | AC | 0.157 | 15.5 | 100.4 | 389.7 | 0.0093 | 0.0093 | 0.0092 | 0.0093 | 0.25 |
| | Baseline | 0.067 | 34.34 | 30783 | 822.3 | 0.0089 | 0.00924 | 0.00914 | 0.0092 | 100 |
| Superconductivity 60 | MR | 2.541 | 33.24 | 30659 | 828.62 | 0.0089 | 0.01027 | 0.00924 | 0.0092 | 75.00 |
| | AMR | 80.40 | 84.26 | 157.54 | 100.12 | 0.0121 | 0.0121 | 0.0121 | 0.0121 | 0.03 |
| | AC | 0.20 | 14.42 | 115.6 | 266.51 | 0.0093 | 0.0093 | 0.0093 | 0.00935 | 0.25 |
| | Baseline | 0.0681 | 34.41 | 30637 | 838.62 | 0.0089 | 0.0104 | 0.00924 | 0.0092 | 100 |
| Gas 20 | MR | 0.566 | 35.05 | 4160 | 577.8 | 0.1079 | 0.1069 | 0.1098 | 0.1043 | 65 |
| | AMR | 1.16 | 4.59 | 31.21 | 11.25 | 0.315 | 0.315 | 0.315 | 0.315 | 0.03 |
| | AC | 0.144 | 20.15 | 108.52 | 228.41 | 0.1069 | 0.1072 | 0.1071 | 0.1065 | 2.01 |
| | Baseline | 0.4472 | 38.67 | 4267 | 600.3 | 0.1056 | 0.1073 | 0.1096 | 0.1063 | 100 |
| Gas 40 | MR | 0.781 | 4096 | 35.8 | 591.21 | 0.1121 | 0.1161 | 0.1101 | 0.1066 | 65.00 |
| | AMR | 1.90 | 5.14 | 34.52 | 10.89 | 0.387 | 0.387 | 0.387 | 0.387 | 0.02 |
| | AC | 0.163 | 16.83 | 125.6 | 163.7 | 0.101 | 0.102 | 0.1065 | 0.11 | 2.01 |
| | Baseline | 0.447 | 38.67 | 5227 | 579.42 | 0.1047 | 0.1045 | 0.1054 | 0.1057 | 100 |
| Gas 60 | MR | 0.566 | 35.05 | 5098 | 283.75 | 0.1983 | 0.1626 | 0.1166 | 0.1536 | 65.00 |
| | AMR | 1.60 | 4.94 | 38.45 | 15.46 | 0.301 | 0.301 | 0.301 | 0.301 | 0.01 |
| | AC | 0.158 | 16.9 | 101.2 | 154.3 | 0.248 | 0.228 | 0.221 | 0.222 | 1.78 |
| | Baseline | 0.447 | 38.67 | 5227 | 283.29 | 0.185 | 0.192 | 0.2001 | 0.198 | 100 |
| Concrete 20 | MR | 0.030 | 0.0501 | 269 | 1.17 | 0.015 | 0.0152 | 0.0155 | 0.015 | 50.00 |
| | AMR | 0.38 | 0.38 | 5.12 | 0.49 | 0.0564 | 0.0564 | 0.0564 | 0.0564 | 0.05 |
| | AC | 0.01 | 0.02 | 23.5 | 0.495 | 0.0153 | 0.01594 | 0.0160 | 0.159 | 6.89 |
| | Baseline | 0.0175 | 0.0627 | 273 | 1.35 | 0.015 | 0.0152 | 0.0156 | 0.015 | 100 |
| Concrete 40 | MR | 0.0383 | 0.0501 | 532 | 1.22 | 0.015 | 0.0147 | 0.0161 | 0.0149 | 50.00 |
| | AMR | 0.74 | 0.72 | 6.12 | 0.85 | 0.070 | 0.070 | 0.070 | 0.070 | 0.03 |
| | AC | 0.012 | 0.024 | 35.4 | 0.511 | 0.0153 | 0.0159 | 0.0159 | 0.0159 | 5.63 |
| | Baseline | 0.0238 | 0.0582 | 536 | 1.489 | 0.015 | 0.0162 | 0.0161 | 0.015 | 100 |
| Concrete 60 | MR | 0.0409 | 0.0504 | 715 | 1.208 | 0.0151 | 0.0148 | 0.0164 | 0.0151 | 50.00 |
| | AMR | 0.96 | 0.96 | 11.25 | 1.08 | 0.0819 | 0.0819 | 0.0819 | 0.0918 | 0.02 |
| | AC | 0.0106 | 0.024 | 46.8 | 0.51 | 0.154 | 0.0163 | 0.0162 | 0.0162 | 5.28 |
| | Baseline | 0.0319 | 0.0561 | 724 | 1.42 | 0.0151 | 0.0153 | 0.0168 | 0.015 | 100 |

As mentioned in the main content, the time complexity of the algorithm is $\mathcal{O}(T_{train} \cdot |MVF(\mathbf{z})|)$, making it significantly more efficient than the baseline algorithm, which trains models over all repairs individually and has a time complexity of $\mathcal{O}(T_{train} \cdot |\mathbf{X}^R|)$. If a gradient descent algorithm is used, Algorithm A has a time complexity of $\mathcal{O}(n \cdot d^3)$, where $n$ is the number of training samples and $d$ is the number of features. In cases where $n < d^2$, the time complexity can be reduced to

Table F: Baseline and optimized average RAM usage (MB) across all SVM MCAR datasets using KNN, MICE, and RF imputations.

| Data Set | Missingness | KNN (MB) | MICE (MB) | RF (MB) |
|---|---|---|---|---|
| Malware 20 | MR | $1117.88 \pm 32.26$ | – | $1590.53 \pm 26.32$ |
| | Baseline | 948.41 | – | 1553.54 |
| Malware 40 | MR | $1062.91 \pm 6.64$ | – | $1513.21 \pm 17.08$ |
| | Baseline | 1361.96 | – | 1533.32 |
| Malware 60 | MR | $980.17 \pm 8.10$ | – | $1325.63 \pm 8.14$ |
| | Baseline | 1427.75 | – | 1336.04 |
| Tuadromd 20 | MR | $231.82 \pm 5.39$ | $449.32 \pm 3.40$ | $250.21 \pm 5.11$ |
| | Baseline | 256.53 | 438.91 | 237.93 |
| Tuadromd 40 | MR | $252.97 \pm 0.00$ | $442.13 \pm 0.63$ | $258.38 \pm 4.29$ |
| | Baseline | 356.64 | 448.42 | 249.85 |
| Tuadromd 60 | MR | $248.74 \pm 0.00$ | $431.94 \pm 1.04$ | $261.36 \pm 0.01$ |
| | Baseline | 410.12 | 450.02 | 265.16 |
| Default 20 | MR | $1104.84 \pm 7.56$ | $201.51 \pm 2.95$ | $518.21 \pm 5.27$ |
| | Baseline | 3351.83 | 201.60 | 520.94 |
| Default 40 | MR | $1620.02 \pm 0.19$ | $202.02 \pm 0.08$ | $483.82 \pm 2.21$ |
| | Baseline | 3502.82 | 211.12 | 510.90 |
| Default 60 | MR | $1795.03 \pm 0.00$ | $196.68 \pm 0.06$ | $455.88 \pm 5.02$ |
| | Baseline | 3094.94 | 210.98 | 537.61 |

Table G: Baseline and optimized average RAM usage (MB) across all SVM MAR datasets using KNN, MICE, and RF imputations.

| Data Set | Missingness | KNN (MB) | MICE (MB) | RF (MB) |
|---|---|---|---|---|
| Malware 20 | MR | $993.87 \pm 126.31$ | – | $2007.92 \pm 3.40$ |
| | Baseline | 994.18 | – | 2188.73 |
| Malware 40 | MR | $917.31 \pm 115.00$ | – | $2021.64 \pm 5.70$ |
| | Baseline | 1065.12 | – | 2187.40 |
| Malware 60 | MR | $816.95 \pm 163.75$ | – | $1772.30 \pm 17.17$ |
| | Baseline | 1139.50 | – | 2203.27 |
| Tuadromd 20 | MR | $209.48 \pm 15.13$ | $433.15 \pm 2.01$ | $229.71 \pm 0.62$ |
| | Baseline | 261.24 | 439.85 | 240.76 |
| Tuadromd 40 | MR | $203.47 \pm 12.71$ | $420.14 \pm 0.06$ | $215.76 \pm 0.05$ |
| | Baseline | 315.21 | 437.52 | 239.30 |
| Tuadromd 60 | MR | $195.01 \pm 15.02$ | $407.45 \pm 0.31$ | $202.39 \pm 0.74$ |
| | Baseline | 363.12 | 436.17 | 238.28 |
| Default 20 | MR | $1088.75 \pm 11.70$ | $198.16 \pm 2.74$ | $500.35 \pm 2.40$ |
| | Baseline | 3351.95 | 203.22 | 526.95 |
| Default 40 | MR | $1586.52 \pm 14.87$ | $186.76 \pm 0.99$ | $458.04 \pm 1.57$ |
| | Baseline | 3460.50 | 203.51 | 506.00 |
| Default 60 | MR | $1743.08 \pm 14.18$ | $183.20 \pm 0.24$ | $431.29 \pm 1.56$ |
| | Baseline | 3050.28 | 201.71 | 507.08 |

$\mathcal{O}(n \cdot d^2 + n^2 \cdot d)$ under certain conditions by applying incremental learning techniques based on the Sherman-Morrison formula, as outlined below.

### OPTIMIZATION FOR ALGORITHM A

The primary time cost in Algorithm A arises from the need to completely retrain the linear regression model each time a new imputed feature is added to the feature set. This retraining leads to a time complexity of $\mathcal{O}(n \cdot d^3)$ for the algorithm. To address this inefficiency, we propose an optimization

Table I: Baseline and optimized average RAM usage (MB) across all SVM MNAR datasets using KNN, MICE, and RF imputations.

| Data Set | Missingness | KNN (MB) | MICE (MB) | RF (MB) |
|---|---|---|---|---|
| Malware 20 | MR | $917.23 \pm 144.43$ | – | $1987.47 \pm 19.47$ |
| | Baseline | 1060.02 | – | 2205.59 |
| Malware 40 | MR | $801.97 \pm 149.17$ | – | $1717.80 \pm 11.01$ |
| | Baseline | 1167.60 | – | 2176.25 |
| Malware 60 | MR | $847.54 \pm 182.56$ | – | $1502.17 \pm 0.11$ |
| | Baseline | 1254.33 | – | 2202.88 |
| Tuadromd 20 | MR | $207.31 \pm 15.65$ | $423.10 \pm 1.04$ | $216.53 \pm 0.16$ |
| | Baseline | 286.32 | 439.73 | 240.33 |
| Tuadromd 40 | MR | $195.30 \pm 15.87$ | $407.96 \pm 1.19$ | $196.51 \pm 0.23$ |
| | Baseline | 364.70 | 436.82 | 237.68 |
| Tuadromd 60 | MR | $189.19 \pm 21.07$ | $395.02 \pm 2.05$ | $182.25 \pm 0.13$ |
| | Baseline | 412.00 | 438.30 | 238.52 |
| Default 20 | MR | $1158.21 \pm 1.98$ | $186.43 \pm 3.31$ | $484.08 \pm 0.08$ |
| | Baseline | 2652.80 | 199.54 | 526.40 |
| Default 40 | MR | $1446.96 \pm 8.63$ | $182.10 \pm 0.05$ | $412.10 \pm 0.89$ |
| | Baseline | 2638.18 | 205.38 | 507.42 |
| Default 60 | MR | $1317.93 \pm 182.56$ | $176.42 \pm 1.03$ | $372.49 \pm 0.18$ |
| | Baseline | 2949.70 | 205.37 | 500.22 |

---

**Algorithm A** Approximating minimal repair for linear regression efficiently

---

$S_{min} \leftarrow [\quad]$
$MVF(\mathbf{z}) \leftarrow$ set of incomplete features
$Complete(\mathbf{z}) \leftarrow$ set of complete features
$\mathbf{r} \leftarrow LR(Complete(\mathbf{z}), \mathbf{y})$ {The residue vector from performing linear regression between complete features and label}
$\epsilon \leftarrow$ a user-defined threshold for stopping condition
$MaxCosSim \leftarrow \max_{\mathbf{z} \in MVF(\mathbf{z})} |cos(\mathbf{z}, \mathbf{r})|$
**while** $MaxCosSim \leq \epsilon$ **do**
$\quad S_{min} \leftarrow S_{min}.add(\arg\max_{\mathbf{z} \in MVF(\mathbf{z})} |cos(\mathbf{z}, \mathbf{r})|)$
$\quad \mathbf{r} \leftarrow LR(Complete(\mathbf{z}) \cup S_{min}, \mathbf{y})$
$\quad MaxCosSim \leftarrow \max_{\mathbf{z} \in MVF(\mathbf{z})} |cos(\mathbf{z}, \mathbf{r})|$
**end while**
$res \leftarrow S_{min}$

---

using the Sherman-Morrison formula to update the inverse of the feature matrix incrementally Angioli et al. (2025). This method reduces the time complexity of including one new feature to $\mathcal{O}(n^2)$. Consequently, when $n < d^2$, this optimization results in significant time savings.

Given a feature matrix $\mathbf{X}$, a label vector $\mathbf{y}$, and the coefficients $\mathbf{w}$ of the current linear regression model, our objective is to efficiently update $\mathbf{w}$ to incorporate a newly imputed feature vector $\mathbf{x}_{\text{new}}$ into $\mathbf{X}$, forming an updated feature matrix $\mathbf{X}'$, without the necessity of full retraining. When this new feature vector $\mathbf{x}_{\text{new}}$ is added to $\mathbf{X}$, it modifies the original matrix product $\mathbf{X}^T\mathbf{X}$ to $\mathbf{X}^T\mathbf{X} + \mathbf{x}_{\text{new}}\mathbf{x}_{\text{new}}^T$. Applying the Sherman-Morrison formula, the updated inverse of $\mathbf{X}'^T\mathbf{X}'$ (assuming $\mathbf{X}'^T\mathbf{X}'$ is invertible) is given by:

$$(\mathbf{X}'^T\mathbf{X}')^{-1} = (\mathbf{X}^T\mathbf{X})^{-1} - \frac{(\mathbf{X}^T\mathbf{X})^{-1}\mathbf{x}_{\text{new}}\mathbf{x}_{\text{new}}^T(\mathbf{X}^T\mathbf{X})^{-1}}{1 + \mathbf{x}_{\text{new}}^T(\mathbf{X}^T\mathbf{X})^{-1}\mathbf{x}_{\text{new}}} \tag{7}$$

This formulation enables the efficient update of the regression coefficients $\mathbf{w}$, requiring only $O(n^2)$ operations. Implementing at most $d$ such updates results in a complexity of $\mathcal{O}(d \cdot n^2)$. Including the initial model training $\mathcal{O}(d^2 \cdot n)$, the total computational complexity is thus reduced to $\mathcal{O}(n \cdot d^2 + n^2 \cdot d)$.

## MINIMAL REPAIR: FEATURE-WISE OR SAMPLE-WISE

For linear SVM, minimal repair (MR) is defined at the sample level—the algorithm returns a set of samples to repair. This is because the method identifies potential support vectors, which are inherently defined based on individual samples.

In contrast, for linear regression, MR is defined at the feature level—the algorithm selects a subset of features to repair. This stems from the interpretation of linear regression as projecting the residual vector onto the feature space. The approach identifies features that do not contribute to minimizing the training loss, given the current regression residual.

## PROOFS

### PROOF FOR THEOREM 1

Prove the theorem by contradiction. Assume that given a training set $(\mathbf{X}, \mathbf{y})$ and a regularization parameter $C$, two minimal repair sets exist ($\mathbf{S}_{min1}(\mathbf{X}, \mathbf{y}, C)$ and $\mathbf{S}_{min2}(\mathbf{X}, \mathbf{y}, C)$). From the definition of minimal repair set, a certain model exists by either imputing all samples in $\mathbf{S}_{min1}(\mathbf{X}, \mathbf{y}, C)$ or $\mathbf{S}_{min2}(\mathbf{X}, \mathbf{y}, C)$, regardless of imputation results. Further, based on the discussion in previous literature Zhen et al. (2024), a certain model exists when none of the incomplete samples is a support vector in any repair. Therefore, if an incomplete sample is not in the minimal repair set, it is not a support vector in any repair. From the assumption, we can always find an incomplete sample $\mathbf{x}_i$ that $\mathbf{x}_i \notin \mathbf{S}_{min1}(\mathbf{X}, \mathbf{y}, C)$ and $\mathbf{x}_i \in \mathbf{S}_{min2}(\mathbf{X}, \mathbf{y}, C)$. In this scenario, $\mathbf{x}_i$ is not a support vector for any repair of $\mathbf{X}$ because $\mathbf{x}_i \notin \mathbf{S}_{min1}(\mathbf{X}, \mathbf{y}, C)$. Thus, $\mathbf{S}_{min2}(\mathbf{X}, \mathbf{y}, C)$ is not a minimal repair set because removing $\mathbf{x}_i$ from $\mathbf{S}_{min1}(\mathbf{X}, \mathbf{y}, C)$ should construct a smaller set also ensuring the existence of certain models, violating the definition of minimal repair set. Contradicting to the original assumption, Theorem 1 holds.

### PROOF FOR LEMMA 1

Borrowing the discussion from proving Theorem 1, if an incomplete sample $\mathbf{x}_i$ is not a support vector in any repair of $\mathbf{X}$, it should not be part of the minimal repair set $S_{min}$ (which is unique from Theorem 1). Further, if an incomplete sample $\mathbf{x}_i$ is a support vector in at least one repair of $\mathbf{X}$, it has to be included in the minimal repair set, otherwise certain model does not exist Zhen et al. (2024).

### PROOF FOR THEOREM 3

Necessity is trivial based on Lemma 2: if an incomplete sample is a support vector in an edge repair, the incomplete sample is part of the minimal repair set. Then we prove sufficiency by contradiction. Assume that there is an incomplete sample $\mathbf{x}_i$ part of the minimal repair set $X_{min}$ while it is not a support vector in any edge repair $\mathbf{x}^e \in \mathbf{X}^E$. Training an SVM can be interpreted as finding the minimal distance between two reduced convex hulls Bennett & Bredensteiner (2000), and if an sample is within the reduced convex hull (not at the boundary), the sample is not a support vector. Because $\mathbf{x}_i$ is not a support vector for any edge repair from the assumption, it is not a support vector for any repair to $\mathbf{X}$. This is because, in the process of changing a value for a missing value ($x_{pq}$) from one edge repair ($x_{pq}^{min}$) to another ($x_{pq}^{max}$) monotonically increase or decrease the coverage of the reduced convex hull. With that being said, if an incomplete sample $\mathbf{x}_i$ is not a support vector for any edge repair (i.e., within the reduced convex hull), the incomplete sample is within the reduced convex hull (i.e., not a support vector) with respect to any repair. This contradicts to the original assumption that $\mathbf{x}_i$ is part of the minimal repair set.

### PROOF FOR THEOREM 4

We reduce from the NP-complete problem 3-SAT. Let

$$\Phi = \bigwedge_{j=1}^{m} (C_j)$$

be a 3-SAT formula with $k$ Boolean variables $z_1, z_2, \ldots, z_k$ and $m$ clauses $C_1, \ldots, C_m$, each clause being a disjunction of three literals.

For each variable $z_\ell$, we introduce one or more *incomplete* samples whose feature vectors each contain a *missing* coordinate $u_\ell$. The imputation set for $u_\ell$ is $\{-1, +1\}$, corresponding to {False, True}. Thus, any assignment of the $z_\ell$ corresponds to choosing $\pm 1$ for these missing coordinates.

To enforce that each clause $C_j$ must be satisfied, we add appropriately labeled points (some possibly incomplete) and arrange them in a geometry so that assigning a literal to *false* yields a large penalty term in the soft-margin objective (either by misclassification or forcing the margin to collapse). Intuitively, if a clause were unsatisfied (all literals set to *false*), the SVM would incur a prohibitively large hinge-loss cost, making that repair suboptimal.

We designate one particular incomplete sample $\mathbf{x}_i$ with additional coordinates or constraints so that:

- *If* $\Phi$ is satisfiable, then there is an imputation (choosing $\pm 1$ consistently with a satisfying assignment) that maximizes the margin while placing $\mathbf{x}_i$ *exactly on* the decision boundary, making it a support vector.

- *If* $\Phi$ is unsatisfiable, then *every* imputation leads to $\mathbf{x}_i$ being off the margin (either strictly inside or otherwise not a support vector). In other words, no selection of $\{\pm 1\}$ for the missing attributes can force $\mathbf{x}_i$ onto the margin.

By suitably tuning the soft-margin parameter $C$ and the placement of the clause-encoding points, we ensure that the SVM will "prefer" to assign $\pm 1$ values in a way that satisfies $\Phi$, whenever possible, in order to avoid a large penalty.

Hence,

$$\Phi \text{ is satisfiable} \iff \text{there exists a repair making } \mathbf{x}_i \text{ a support vector.}$$

Since deciding satisfiability for $\Phi$ (3-SAT) is NP-complete, it follows that deciding whether $\mathbf{x}_i$ can be a support vector under some imputation is NP-hard.

Determining membership of a single incomplete sample $\mathbf{x}_i$ among the possible support vectors is NP-hard. Therefore, listing *all* such samples that can ever appear on the margin is also NP-hard: if we had such a list in polynomial time, we could decide membership in that list in polynomial time, contradicting NP-hardness. Given the proof that finding MR for SVM is NP-hard, deciding whether an incomplete sample belongs to the MR for SVM is also NP hard. To prove, assume that we have a polynomial-time solver for deciding whether an incomplete sample belongs to the MR, then one can linearly scan each incomplete sample and decide its membership in MR (either belongs to or not) by calling the polynomial time subroutine. Therefore, one can find the MR in polynomial time, which contradicts to the NP-hard proof earlier.

PROOF FOR THEOREM 5

For any incomplete sample $\mathbf{x}_i$ returned from Algorithm 1 in main content for SVM, the incomplete sample is a support vector in at least one repair to $\mathbf{X}$. Based on Theorem 3, it is part of the minimal repair.

PROOF FOR THEOREM 6

Given the iterative algorithm of finding the minimal repair for SVM (Algorithm 1 in the main content), we first characterize the probability that the imputation set returned at iteration $k$ misses one or more incomplete samples that belong to the minimal repair.

Let $k$ be the current iteration index ($k = 0$ represents the initial state before the first run). We define the following: $MS(x)^k$ is the set of incomplete samples remaining at the start of iteration $k$. $M^k = |MS(x)^k|$ is the number of remaining incomplete samples at the start of iteration $k$. $S_{min}^k$ is the (unknown) true minimal set of samples within $MS(x)^k$ that must be imputed at the start of iteration $k$ to guarantee a certain model. $s^k = |S_{min}^k|$ is the (unknown) size of this true minimal set; note that we treat $s^k$ as a random variable, and $s^k \leq M^k$. $S'^k$ is the set of samples returned by Algorithm 1 in the main content when run at iteration $k$ on the current data; we know $S'^k \subseteq S_{min}^k$. $FN^k$ is the event that makes at least one false negative error at iteration $k$, occurring if $S'^k$ is a proper

subset of $S_{min}^k$. $P(FN^k)$ is the probability of event $FN^k$. We seek a computable upper bound $UB'^k$ such that $P(FN^k) \leq UB'^k$. define $p_{fn}$ as an upper bound on the per-sample false negative probability, $p(\mathbf{x_i})$. We assume that there exists a probability $p_{fn}$ (where $0 \leq p_{fn} \leq 1$) such that for any sample $x_i \in S_{min}^k$, the probability that Algorithm 1 in the main content fails to include $x_i$ in $S'^k$ is bounded above by $p_{fn}$:

$$P(x_i \notin S'^k | x_i \in S_{min}^k) \leq p_{fn}$$

Then we propose, $UB'^k$, an upper bound of $P(FN^k)$ as follows:

$$UB'^k = 1 - (1 - p_{fn})^{M^k} \geq P(FN^k)$$

To interpret, when the iteration goes (k becomes larger), $M^k$ and $p_{fn}$ decrease (which we will prove later), $UB'^k$ decreases. This indicates that the upper-bound of probability of under-imputing decreases over iterations.

To prove this bound, we begin by expressing the target probability $P(FN^k)$ using its complement. The event $FN^k$ (at least one false negative) is the complement of the event $NoFN^k$ (no false negatives, i.e., $S'^k = S_{min}^k$). Therefore, conditioned on the true size $s^k$ of the minimal set at iteration $k$, we have $P(FN^k | s^k) = 1 - P(\text{No FN}^k | s^k)$.

Next, we bound the probability of having no false negatives, $P(\text{No FN}^k | s^k)$. The event $NoFN^k$ occurs if Algorithm 1 in the main content successfully returns all samples in $S_{min}^k$. Let $E_i$ be the event that Algorithm 1 in the main content fails to return sample $x_i$. Assuming the failure/success events $E_i$ for different samples $x_i \in S_{min}^k$ within the same iteration $k$ are statistically independent, we can write:

$$P(\text{No FN}^k | s^k) = P(\cap_{x_i \in S_{min}^k} \{\text{not } E_i\} | s^k) = \prod_{x_i \in S_{min}^k} P(\text{not } E_i | s^k)$$

Let $P(E_i | s^k)$ be the probability of failure for $x_i$. Then $P(\text{not } E_i | s^k) = 1 - P(E_i | s^k)$. Using the definition $P(E_i | s^k) \leq p_{fn}$, we have $1 - P(E_i | s^k) \geq 1 - p_{fn}$. Substituting this lower bound into the product gives:

$$P(\text{No FN}^k | s^k) \geq \prod_{i=1}^{s^k} (1 - p_{fn}) = (1 - p_{fn})^{s^k}$$

Now we can bound $P(FN^k | s^k)$:

$$P(FN^k | s^k) = 1 - P(\text{No FN}^k | s^k) \leq 1 - (1 - p_{fn})^{s^k}$$

The overall probability $P(FN^k)$ is the expectation over the unknown size $s^k$:

$$P(FN^k) = \mathbb{E}_{s^k}[P(FN^k | s^k)] \leq \mathbb{E}_{s^k}[1 - (1 - p_{fn})^{s^k}]$$

To proceed, we utilize Jensen's inequality. Let $f(s) = 1 - (1 - p_{fn})^s$. We first prove that $f(s)$ is concave for $s \geq 0$. Let $b = 1 - p_{fn}$. Since $0 \leq p_{fn} < 1$, we have $0 < b \leq 1$. The function is $f(s) = 1 - b^s$. The first derivative is $f'(s) = -b^s \ln(b)$. The second derivative is $f''(s) = -(b^s \ln(b)) \ln(b) = -b^s (\ln(b))^2$. Since $b^s > 0$ and $(\ln(b))^2 \geq 0$, the second derivative $f''(s) \leq 0$. Therefore, $f(s)$ is a concave function.

Jensen's inequality for a concave function $f$ states $\mathbb{E}[f(X)] \leq f(\mathbb{E}[X])$. Applying this to our expectation:

$$\mathbb{E}_{s^k}[1 - (1 - p_{fn})^{s^k}] \leq 1 - (1 - p_{fn})^{\mathbb{E}[s^k]}$$

Combining this with the previous inequality gives a theoretical upper bound:

$$P(FN^k) \leq 1 - (1 - p_{fn})^{\mathbb{E}[s^k]}$$

The term $\mathbb{E}[s^k]$ (expected number of truly needed samples) is still unknown. However, we know that the number of needed samples $s^k$ cannot exceed the total number of remaining incomplete samples $M^k = |MS(x)^k|$. Thus, $s^k \leq M^k$. Taking expectations yields $\mathbb{E}[s^k] \leq \mathbb{E}[M^k]$. Since $M^k$ is a

known quantity (computable by counting) at the start of iteration $k$, $\mathbb{E}[M^k] = M^k$. Therefore, we have a computable upper bound for the expectation: $\mathbb{E}[s^k] \leq M^k$.

Finally, we substitute this bound on $\mathbb{E}[s^k]$ into the Jensen result. Let $g(x) = (1 - p_{fn})^x$. Since $0 < (1 - p_{fn}) \leq 1$, $g(x)$ is a non-increasing function. Applying $g$ to the inequality $\mathbb{E}[s^k] \leq M^k$ reverses the inequality direction:

$$(1 - p_{fn})^{\mathbb{E}[s^k]} \geq (1 - p_{fn})^{M^k}$$

Multiplying by -1 and adding 1 (reversing the inequality twice):

$$1 - (1 - p_{fn})^{\mathbb{E}[s^k]} \leq 1 - (1 - p_{fn})^{M^k}$$

Combining the inequalities $P(FN^k) \leq 1 - (1 - p_{fn})^{\mathbb{E}[s^k]}$ and $1 - (1 - p_{fn})^{\mathbb{E}[s^k]} \leq 1 - (1 - p_{fn})^{M^k}$, we arrive at the final upper bound $UB'^k$:

$$P(FN^k) \leq 1 - (1 - p_{fn})^{M^k}$$

and

$$UB'^k = 1 - (1 - p_{fn})^{|MS(x)^k|}$$

Now the only problem is to compute $p_{fn}$ and understand how it changes over iterations. The Multiple Random Starts method provides an empirical approach. First, select a set of incomplete samples $MS_{probe}$ (e.g., $MS(x)^0$) and choose the number of repetitions $T$ (e.g., $T = 10$ or $20$). For each $x_i \in MS_{probe}$, initialize a success count $t_i = 0$. Repeat $T$ times: generate a new random edge repair $X^e_{start,t}$ for the current dataset state; run the greedy construction part of Algorithm 1 in the main content starting from $X^e_{start,t}$ to get $X^e_{final,i,t}$; train $w_{final,i,t} = SVM(X^e_{final,i,t}, y)$; check if $y_i(w_{final,i,t})^T(x_i$ part of $X^e_{final,i,t}) \leq 1$. If yes, increment $t_i$.

Also, if the probability distribution of each incomplete sample is known, and we let $g(x_{ij})$ denote the probability density function of the ground truth value for the missing value $x_{ij}$ in the incomplete training set $(\mathbf{X}, \mathbf{y})$. If missing values in $\mathbf{X}$ are independent, the probability that an incomplete sample $\mathbf{x}_i$ in minimal repair not returned by Algorithm 1 in the main content is:

$$p(\mathbf{x}_i) = 1 - \frac{\int \cdots \int_{\min(x_{ij}^{\text{visited}})}^{\max(x_{ij}^{\text{visited}})} \prod_{x_{ij} \in M(\mathbf{X})} g(x_{ij}) \, dx_{ij}}{\int \cdots \int_{x_{ij} \in M(\mathbf{X})} \prod_{x_{ij} \in M(\mathbf{X})} g(x_{ij}) \, dx_{ij}}$$

$x_{ij}^{\text{visited}} \in \{x_{ij}^{min}, x_{ij}^{max}\}$ shows the values used for $x_{ij}$ in Algorithm 1 in the main content. It shows that the more edge repairs Algorithm 1 explores, the lower the false negative probability for each sample. One can find $p_{fn}$ by computing $p(\mathbf{x}_i)$ for each incomplete sample and take the maximum as $p_{fn}$. $p_{fn}$ decreases over iterations because each iteration explores additional edge repairs. This expands the domain of the numerator in the expression increasing the integral value and thereby lowering $p(\mathbf{x}_i)$ for every sample5. Since $p_{fn}$ is an upper bound over all such $p(\mathbf{x}_i)$, it decreases as well.

PROOF FOR THEOREM 7

Prove the possibility of having multiple minimal repair sets first. Because linear regression can have multiple non-trivial optimal models in general, multiple minimal repair sets can exist, and each multiple imputation set corresponds to an optimal linear regression model. For example, when we have the dataset:

$$X = \begin{bmatrix} 1 & \text{null} & \text{null} & \text{null} \\ 0 & 1 & 2 & 3 \\ 0 & 4 & 3 & 2 \end{bmatrix}, \quad y = \begin{bmatrix} 1 \\ 1 \\ 1 \end{bmatrix}.$$

We denote features from left to right as $\mathbf{z}_1 \ldots \mathbf{z}_4$. In this example, there are at least two MRs, $MR_1 = \{\mathbf{z}_2, \mathbf{z}_3\}$ and $MR_2 = \{\mathbf{z}_3, \mathbf{z}_4\}$. To prove, we first show that imputing either $MR_1$ or $MR_2$, and training a linear regression model with imputed features and the originally complete feature ($\mathbf{z}_1$) leads to a zero (minimal) regression loss in all repairs of $X$. Let us first consider $MR_1$. The two incomplete features ($\mathbf{z}_2$ and $\mathbf{z}_3$) with the complete one ($\mathbf{z}_1$) cover the full 3-dimensional space

in all repairs because the three features are linearly independent in all repairs. We show the linear independence by computing the determinant of the matrix $A$ consisting of $\mathbf{z}_1$, $\mathbf{z}_2$, and $\mathbf{z}_3$.

$$A = \begin{bmatrix} 1 & null & null \\ 0 & 1 & 2 \\ 0 & 4 & 3 \end{bmatrix}$$

The determinant of the matrix $A$ is non-zero regardless of how the null values in $\mathbf{z}_2$ and $\mathbf{z}_3$ are imputed.

$$\det(A) = \det(A^T) = 1 \cdot \det \begin{pmatrix} 1 & 4 \\ 2 & 3 \end{pmatrix} - 0 \cdot \det \begin{pmatrix} null & 4 \\ null & 3 \end{pmatrix} + 0 \cdot \det \begin{pmatrix} null & 1 \\ null & 2 \end{pmatrix}$$
$$= 1 \cdot ((1)(3) - (2)(4))$$
$$= 1 \cdot (3 - 8)$$
$$= -5$$

Because $\mathbf{z}_1$, $\mathbf{z}_2$, and $\mathbf{z}_3$ are linearly independent, for every repair of $A$, there is a linear regression model that achieves zero (minimal) loss with the feature matrix $A$ and the label vector $y$. Let $v(\mathbf{z}_2)$ and $v(\mathbf{z}_3)$ denote a repair of columns (features) $\mathbf{z}_2$ and $\mathbf{z}_3$ in $A$, respectively. Every repair of the matrix $X$ with $v(\mathbf{z}_2)$ and $v(\mathbf{z}_3)$ for its second and third columns, no matter what the imputation of missing value in $\mathbf{z}_4$ is, will have zero regression loss for the label vector $y$.

Similarly, for $MR_2$, we show that the two incomplete features ($\mathbf{z}_3$ and $\mathbf{z}_4$) along with the complete $\mathbf{z}_1$ cover the full 3-dimensional space in all repairs because the three features are linearly independent in all repairs. We show this by computing the determinant of the matrix $B$ consisting of $\mathbf{z}_1$, $\mathbf{z}_3$, and $\mathbf{z}_4$.

$$B = \begin{bmatrix} 1 & null & null \\ 0 & 2 & 3 \\ 0 & 3 & 2 \end{bmatrix}$$

The determinant of $B$ is non-zero in all repairs.

$$\det(B) = \det(B^T) = 1 \cdot \det \begin{pmatrix} 2 & 3 \\ 3 & 2 \end{pmatrix} - 0 \cdot \det \begin{pmatrix} null & 3 \\ null & 2 \end{pmatrix} + 0 \cdot \det \begin{pmatrix} null & 2 \\ null & 3 \end{pmatrix}$$
$$= 1 \cdot ((2)(2) - (3)(3))$$
$$= 1 \cdot (4 - 9)$$
$$= -5$$

Therefore, similar to our argument for $MR_1$, the regression loss for every repair of the features of $MR_2$ in the linear regression with feature matrix $X$ and label vector $y$ is zero (minimal) no matter what the imputation of the missing value in $\mathbf{z}_2$ is.

To close the proof for $MR_1$ and $MR_2$ being minimal repairs, we also show that there is no smaller subset (with only one incomplete feature) such that by imputing the subset and training a linear regression model with the imputed feature and the originally complete feature $\mathbf{z}_1$ leads to the minimal regression loss in all repairs. By scanning every single incomplete feature, no one can achieve the minimal regression loss along with the complete feature ($\mathbf{z}_1$) in all repairs. Therefore, the size of MR should be 2, which concludes the proof that $MR_1$ and $MR_2$ are both minimal repairs in this example dataset. However, when all features in $\mathbf{X}$ are linearly independent in all repairs, the optimal linear regression model is unique for every repair. Therefore, a certain model is unique when it exists in this scenario, and the minimal repair set is also unique to reach a certain model.

PROOF FOR THEOREM 8

To prove that finding the linear regression solution that is most sparse over a subset of features is NP-hard, we reduce the known NP-hard problem of finding the most sparse linear regression solution

to it Bruckstein et al. (2009). Consider the original problem where given a feature matrix $\mathbf{X}$ and a label vector $\mathbf{y}$, the goal is to find the optimal model $\mathbf{w}^*$ that minimizes the number of non-zero entries. In the new problem, given a subset of features, i.e., the incomplete features, denoted as $MVF(\mathbf{X})$, we seek the optimal model $\mathbf{w}^*$ that minimizes the number of non-zero entries in the coefficients within $MVF(\mathbf{X})$. To reduce the original problem to this new one, set $MVF(\mathbf{X})$ as the entire feature set. Solving the new problem in this special case is equivalent to solving the original sparse linear regression problem, which is NP-hard. Therefore, the new problem must also be NP-hard, as it generalizes the original problem.

PROOF FOR LEMMA 9

Based on the previous literature about certain model Zhen et al. (2024), when a certain model $\mathbf{w}^*$ exists for linear regression, $w_i = 0$ for every $\mathbf{z}_i \in MVF(\mathbf{X})$. Therefore, finding a minimal repair set in linear regression is equivalent to finding a regression model that has the maximal number of zero model parameters (linear coefficients) and is optimal for all repairs. Further, the problem is equivalent to minimizing the number of non-zero linear coefficients in $\mathbf{w}$ whose corresponding feature is incomplete.

PROOF FOR THEOREM 10

When each missing value in the dataset follows an independent zero-mean normal distribution, training a linear regression model based on the incomplete dataset is equivalent to training linear regression with a zero-mean Gaussian noise $\epsilon$ as below:

$$\mathbf{y} = \mathbf{X}\mathbf{w} + \epsilon$$

Based on previous literature Cai & Wang (2011), in the presence of a Gaussian noise $\epsilon \sim \mathcal{N}(0, \sigma^2)$, the first $k$ features returned from OMP method is correct with a probability of at least $1 - 1/n$ when the following two conditions are satisfied: 1. $\mu < 1/(2k-1)$, and 2.

$$|w_i| \geq \frac{2\sigma_{ij}\sqrt{n + 2\sqrt{n\log n}}}{1 - (2k-1)\mu}$$

As a result, the features returned by the OMP algorithm in our paper is correct with a probability of at least $1 - 1/n$ given the conditions in Theorem 10.

PROOF FOR THEOREM 11

The proof has two parts: (1) showing that any set of samples $S'_k$ selected by ST2 at iteration $k$ is a subset of $S_{\text{AMR}}$, implying $S_{\text{iter-ACM}} = \bigcup_k S'_k \subseteq S_{\text{AMR}}$; and (2) showing the algorithm terminates with an ACM ($g_k \leq e$).

**Part 1: Each selection $S'_k$ by ST2 belongs to $S_{\text{AMR}}$**

$S_{\text{AMR}}$ is the smallest set of incomplete samples in $\mathbf{X}$ whose robust imputation guarantees $g \leq e$, irrespective of specific repair values. Consider iteration $k$: ST1 operates on $\mathbf{X}^{(k)}$ (where $S_{\text{iter-ACM}}^{(k-1)} = \bigcup_{i<k} S'_i$ are imputed) yielding $g_k$. If $g_k > e$, ST2 returns $S'_k$, the minimal set of currently incomplete samples in $\mathbf{X}^{(k)}$ necessary to enable $g < g_k$ in the next iteration.

Let $x_j \in S'_k$. Assume, for contradiction, $x_j \notin S_{\text{AMR}}$. If $x_j \notin S_{\text{AMR}}$, then $S_{\text{AMR}}$ (not containing $x_j$) robustly guarantees $g \leq e$ for the original problem $(\mathbf{X}, \mathbf{y})$. So, $x_j$ is not required for this global robust guarantee. At iteration $k$, ST2 identifies $x_j$ as part of the minimal set $S'_k$ in $\mathbf{X}^{(k)}$ needed to reduce $g_k$. This implies $x_j$ is locally indispensable for progress from $\mathbf{X}^{(k)}$.

Let $S^*_{\text{AMR}} = S_{\text{AMR}} \cap U^{(k)}$ be the $S_{\text{AMR}}$ samples still incomplete in $\mathbf{X}^{(k)}$. By induction ($S_{\text{iter-ACM}}^{(0)} = \emptyset \subseteq S_{\text{AMR}}$), all $S_{\text{iter-ACM}}^{(k-1)} \subseteq S_{\text{AMR}}$. If $S_{\text{AMR}}$ (excluding $x_j$) robustly guarantees ACM for $\mathbf{X}$, and $S_{\text{iter-ACM}}^{(k-1)} \subseteq S_{\text{AMR}}$, then any local impasse $g_k > e$ must be resolvable by further imputing only samples from $S^*_{\text{AMR}}$. So, some $P \subseteq S^*_{\text{AMR}}$ must exist to allow $g$ to decrease. Since ST2 returns the *minimal* set for progress, if such $P$ exists, ST2 would select $S'_k \subseteq P \subseteq S^*_{\text{AMR}} \subseteq S_{\text{AMR}}$. This means $x_j \in S_{\text{AMR}}$, contradicting $x_j \notin S_{\text{AMR}}$.

Thus, if ST2 selects $x_j$ (assumed $x_j \notin S_{\text{AMR}}$) as part of $S'_k$, it means no $P \subseteq S^*_{\text{AMR}}$ alone allows progress, and $x_j$ is also needed. This implies $x_j$ is locally indispensable even if all of $S^*_{\text{AMR}}$ were imputed. This contradicts the global sufficiency of $S_{\text{AMR}}$ (which excludes $x_j$). The perfection of ST2 ensures it doesn't select a globally redundant $x_j$ if progress is possible via samples in $S^*_{\text{AMR}}$. So, $x_j \notin S_{\text{AMR}}$ is false. Thus, any $x_j \in S'_k$ is in $S_{\text{AMR}}$, meaning $S'_k \subseteq S_{\text{AMR}}$ for all $k$. Consequently, $S_{\text{iter-ACM}} = \bigcup_k S'_k \subseteq S_{\text{AMR}}$.

**Part 2: Algorithm Termination with an ACM**

If $g_k > e$, ST2 identifies a non-empty $S'_k$ for imputation. (If $S'_k$ was empty while $g_k > e$, it would contradict the existence of $S_{\text{AMR}}$ as a solution or the ideal functioning of ST1/ST2). Imputing $S'_k$ creates $\mathbf{X}^{(k+1)}$. The number of incomplete samples is finite. ST2 selects un-imputed samples necessary for reducing $g_k$. Assuming perfect ST1/ST2, the algorithm progresses towards $g_k \leq e$. It cannot impute distinct samples indefinitely nor cycle with $g_k > e$ as each ST2 selection resolves a current bottleneck. Thus, it must reach $g_k \leq e$ and terminate, achieving ACM.

The assertion $S_{\text{iter-ACM}} \subset S_{\text{AMR}}$ is consistent: $S_{\text{AMR}}$ ensures robustness for *all* repairs. The algorithm uses specific repairs and may achieve ACM before all of $S_{\text{AMR}}$ (needed for worst-case robustness) are imputed.

PROOF FOR THEOREM 12

We assume that the loss function $L(\mathbf{w})$ is convex and has an $M$-Lipschitz continuous gradient. Formally, this means for all $\mathbf{w}, \mathbf{w}' \in \mathcal{W}$:

$$\|\nabla L(\mathbf{w}) - \nabla L(\mathbf{w}')\| \leq M\|\mathbf{w} - \mathbf{w}'\|.$$

Since $L(\mathbf{w})$ is convex with an $M$-Lipschitz continuous gradient, the following standard inequality from convex optimization theory holds:

$$L(\mathbf{w}) \leq L(\mathbf{w}') + \nabla L(\mathbf{w}')^\top (\mathbf{w} - \mathbf{w}') + \frac{M}{2}\|\mathbf{w} - \mathbf{w}'\|^2, \quad \forall \mathbf{w}, \mathbf{w}' \in \mathcal{W}.$$

Let $\mathbf{w}^*$ be an optimal solution (thus $\nabla L(\mathbf{w}^*) = 0$), and set $\mathbf{w}' = \mathbf{w}^*$, then we have:

$$L(\mathbf{w}^\approx) \leq L(\mathbf{w}^*) + \frac{M}{2}\|\mathbf{w}^\approx - \mathbf{w}^*\|^2.$$

Next, due to convexity of $L(\mathbf{w})$, we have:

$$L(\mathbf{w}^*) \geq L(\mathbf{w}^\approx) + \nabla L(\mathbf{w}^\approx)^\top (\mathbf{w}^* - \mathbf{w}^\approx).$$

Combining the two inequalities, we get:

$$L(\mathbf{w}^\approx) - L(\mathbf{w}^*) \leq \frac{M}{2}\|\mathbf{w}^\approx - \mathbf{w}^*\|^2 \leq \frac{1}{2M}\|\nabla L(\mathbf{w}^\approx)\|^2,$$

where the last step follows from the Lipschitz continuity of the gradient, which implies that:

$$\|\nabla L(\mathbf{w}^\approx)\| \geq M\|\mathbf{w}^\approx - \mathbf{w}^*\|.$$

Hence, the optimality gap is explicitly bounded by the norm of the gradient:

$$L(\mathbf{w}^\approx) - L(\mathbf{w}^*) \leq \frac{1}{2M}\|\nabla L(\mathbf{w}^\approx)\|^2.$$

Therefore, to guarantee for all $\mathbf{X}^r \in \mathbf{X}^R$ that:

$$L(f(\mathbf{X}^r, \mathbf{w}^\approx), \mathbf{y}) - \min_{\mathbf{w} \in \mathcal{W}} L(f(\mathbf{X}^r, \mathbf{w}), \mathbf{y}) \leq e,$$

it is sufficient to require:

$$\|\nabla_{\mathbf{w}} L(f(\mathbf{X}^r, \mathbf{w}^{\approx}), \mathbf{y})\| \leq \sqrt{2Me}, \quad \forall \mathbf{X}^r \in \mathbf{X}^R.$$

This completes the derivation.

## ADDITIONAL RELATED WORK

There are methods to detect cases where the imputation of missing data is not necessary to learn accurate models Picado et al. (2020); Karlaš et al. (2020); Zhen et al. (2024). Although these approaches are useful for some datasets and learning tasks, they ignore a majority of learning tasks in which imputing incomplete samples affects the quality of the learned model.

Researchers have proposed methods to reduce the cost of repair Krishnan et al. (2016); Karlaš et al. (2020). ActiveClean learns models using stochastic gradient descent and greedily chooses samples for repair that may reduce the gradient the most Krishnan et al. (2016). Unlike our methods, it does not provide any guarantees of minimal repair. Due to the inherent properties of stochastic gradient descent, it is challenging to provide such a guarantee. CPClean follows a similar greedy approach, but is limited to learning k nearest neighbor models over missing data and does not support the types of model our approach addresses Karlaš et al. (2020). It also does not provide any guarantees of minimality for its imputations.

## CODE REPOSITORY

Link: https://anonymous.4open.science/r/Submission_2025-A1C0/README.md

