# OpenReview forum: "Minimal Repairs for Learning Over Incomplete Data"
_ICLR.cc/2026/Conference — Submitted to ICLR 2026_

### Official Review · Reviewer_eEJo · 2025-10-18

**Soundness:** 3
**Presentation:** 3
**Contribution:** 1
**Rating:** 2
**Confidence:** 4

**Summary:**

The paper defines Minimal Repair (MR) and Almost Minimal Repair (AMR) for learning with incomplete data. The authors give NP‑hardness proofs for MR/AMR with SVMs and linear regression, propose approximation algorithms (for SVM via “edge repairs” and support‑vector tests; for linear regression via an OMP‑style procedure), and analyze limited correctness properties (e.g., “no false positives” for the SVM routine; probabilistic inclusion conditions for the LR routine under incoherence and Gaussian assumptions). The paper acknowledges that guarantees and tractability are limited to convex settings and that MR/AMR can take as long as or longer than simple full imputation.

**Strengths:**

S1. Clear formalization of MR/AMR and useful reductions/NP‑hardness results for SVM and linear regression.

S2. Concrete approximation algorithms with some correctness properties (e.g., SVM “no false positives”; LR OMP‑style procedure).

S3. Broad experimental suite across multiple datasets with both injected and naturally occurring missingness.

**Weaknesses:**

W1. “Minimal repair” is decoupled from ground truth; guarantees are model‑relative, not reality‑relative.
MR is defined to ensure existence of a certain model after imputing a subset; the paper explicitly states that this is orthogonal to the accuracy of the imputations and that a certain model may be accurate or inaccurate depending on the imputed values. Thus MR says nothing about recovering true data values or improving true predictive accuracy; it only targets invariance of the ERM across completions. This disconnect undermines the practical value of the theory.

W2. Error bounds are measured w.r.t. MR/ACM, not the true data or generalization performance.
All guarantees bound differences to the minimal (or almost minimal) set or to an approximately certain model—not to the ground‑truth dataset or population risk. For LR, the probabilistic inclusion guarantee relies on mutual incoherence and independent zero‑mean Gaussian assumptions over missing entries—conditions that are rarely realistic—and still only certifies proximity to the MR set, not to the true data or Bayes model. For AMR, the optimizer minimizes a worst‑case suboptimality proxy via sampled edge repairs and gradient/subgradient norms; again, this measures closeness to ACM, not to truth or test performance. Consequently, there is no reason to believe MR is better than AMR in practice, since both are benchmarked against MR/ACM rather than ground truth.

W3. Missing—and arguably more relevant—baselines.
Despite framing the work as practical, the experiments do not compare to LLM‑based repair strategies or other modern learned imputers that can leverage semantics or context (even if guarantee‑free). The baselines listed in the tables are KNN, MICE, TCSDI (diffusion), MF, and ActiveClean; no LLM‑based repair is evaluated, and even some cited learned imputers (e.g., GAIN) are not included empirically. If MR/AMR are meant to reduce human/compute costs while maintaining model utility, comparisons against strong heuristics that practitioners actually use are essential.

W4. Practical relevance is questionable given the paper’s own limitations.
The paper concedes (i) convexity is required for current guarantees/approximations, leaving non‑convex and deep models out of scope, and (ii) MR/AMR can match or exceed the time of full imputation when simple methods are used. In many realistic pipelines, simple imputers or domain‑specific heuristics are exactly what practitioners deploy; if MR/AMR provides no accuracy gains relative to ground truth and may cost as much or more time, its advantage is unclear.

W5. Reliance on restrictive constructs and strong assumptions.
The SVM routine hinges on edge repairs (min/max bounds for every missing value) and support‑vector tests; this presupposes credible per‑feature bounds, which are often unavailable or uninformative for continuous attributes. For LR, the OMP‑style analysis assumes incoherence and Gaussian missing‑value distributions, which may not hold (especially under MNAR). These choices further limit applicability.

W6. Scope limited to SVM/linear regression; no path to deep models.
The framework’s theoretical underpinnings and efficiency claims currently do not extend to non‑convex learners; the paper notes this as an open challenge. For a venue like ICLR, where deep models are central, the lack of a credible path or partial results for non‑convex settings weakens the contribution.

**Questions:**

See the above weaknesses W1-W5.

---

> ### Author Response · Authors · 2025-12-01
>
> **W1**
>
> We use ground-truth (GT) values in our empirical studies (Tables 4, 6, and new tables in our response to Reviewer 3xJ4) to evaluate accuracy. Results indicate our methods learn accurate models by repairing a significantly smaller subset of the data than full imputation with GT values.
> Our goal is to reduce the cost of using *current* data imputation methods (e.g., manual verification), not to develop a new imputation method.
>
> Moreover, our framework can be easily extended to incorporate GT. Given an imputation method's average error rate, we model the induced uncertainty in the imputed values as a range of values. Then, we use Theorem 3 (and Theorem 12 for AMR) to efficiently return a small set of possible models by considering only the *edge repairs* of the uncertain data. We will include these methods in the revised submission.
>
> **W2**
>
> As explained in W1, we benchmark against GT. Regarding theoretical assumptions (e.g., **independent zero-mean Gaussian** for LR): our extensive evaluation indicates that in cases where these do not hold, our approach delivers accurate results. For example, our experiments use datasets with **MNAR** data (e.g., **Air Quality**, **Bankruptcy**, **Online-Ed**) and non-Gaussian distributions (e.g., **Concrete**), which violate these ideal theoretical conditions (shown in Table 6).
>
> Finally, regarding model quality, **MR** indeed delivers more accurate models than **AMR** in practice. Theoretically, **MR** guarantees recovering the **Certain Model** (strict optimality), whereas **AMR** guarantees a model within $\epsilon$-optimality (ACM). Empirically (Tables 3-5), MR achieves higher accuracy compared to AMR. Thus, **MR** is designed for users who prioritize obtaining the **highest possible model accuracy**, whereas **AMR** targets an **approximate** solution to significantly reduce the computational cost of finding the repair and impute.
>
> **W3**
>
> We used representative methods: statistical (**MICE**), traditional ML-based (**KNN**, **Missing Forest**), and deep generative (**TabCSDI**). **TabCSDI** has been shown to outperform earlier deep learning methods like GAIN [*Zheng & Charoenphakdee, Diffusion models for missing value imputation in tabular data, NeurIPS 2022*]. Similarly, surveys indicate that **MICE** remains a competitive standard and widely used imputation method in real-world applications [*Perini & Nikolic, In-Database Data Imputation, SIGMOD 2024 and the references therein*]. We will include these in the revision.
>
> We agree **LLM-based repair** is relevant and will evaluate it in the revision. Regarding cost, we clarify that "manual cleaning" implies verification by **domain experts**. Even against strong heuristics, minimizing the volume of data requiring expensive domain expert effort remains an important benefit.
>
> **W4**
>
> While we proposed MR algorithms only for SVM/LR, our method for **AMR** applies to all convex models. In the revision, we will outline the path for **DNNs**. We leverage the strong representation power of DNNs [*Zhang et al., Understanding deep learning requires rethinking generalization, ICLR 2017*], which results in negligible minimum loss across repairs. This simplifies the min-max optimization steps (ST1 and ST2) for DNNs: our algorithm iteratively identifies the subset violating the worst-case threshold (similar to ST2) and updates the model via SGD (similar to ST1), dynamically delivering the ACM guarantee.
>
> Simple methods (KNN) are fast but often inaccurate. High-stakes applications (e.g., medical) need accurate models [*Yakout et al., Guided data repair, VLDB 2011*], often requiring expensive **manual verification** or complex models. Full imputation with these is often infeasible. MR/AMR makes them feasible: in **Table 4**, full imputation with TabCSDI is more than **20x** slower than the total time of finding and imputing the AMR subset.
>
> **W5**
>
> Feature bounds are approximated using observed min/max; experiments show this results in accurate models.
> Theoretical assumptions (Gaussian) are not required for execution. We demonstrate robustness on **MNAR** and non-Gaussian datasets (Table 6).
> Please refer to our response to **Reviewer 3xJ4** for full MNAR/MAR results. For **Malware-MAR**, we found **TabCSDI** ran for >2 days (due to repeated high-dimensional forward passes), and **MICE** was infeasible (exhausting RAM due to large design matrices). Our algorithm successfully found repairs.
> Similarly, for **SVM** (which lacks smoothness), we utilize subgradient norms as a practical proxy. Our empirical results confirm that these design choices remain robust and effective (Table 4).
>
> **W6**
>
> Please refer to our response to W4.

---

### Official Review · Reviewer_3xJ4 · 2025-10-31

**Soundness:** 2
**Presentation:** 2
**Contribution:** 2
**Rating:** 4
**Confidence:** 3

**Summary:**

This paper utilizes an efficient approximation algorithm to solve the problem of reducing the time and effort needed to train accurate ML models on incomplete data by only imputing necessary missing values. Specifically, it first proposes the concepts of "minimal" and "almost minimal repair" (the minimal subsets of missing data needed for accurate models). Then show finding these sets is NP-hard for SVM/linear regression and propose efficient approximation algorithms to find them. The experimental results prove the superiority of the proposed method by showing a substantial reduction in time and effort required for learning.

**Strengths:**

1)The paper formally defines minimal repair for SVM and linear regression. They prove that finding these repairs is NP-hard, establishing the theoretical difficulty of the problem, and consequently propose efficient approximation algorithms with provable error bounds to make the solution practical.

2)This paper introduces the concept of almost minimal repair, a solution that is easier to find and guarantees a model loss close to that of a fully repaired dataset. They provide the necessary NP-hard proofs and corresponding approximation algorithms for this concept as well.

3)Empirical results demonstrate that the proposed algorithms efficiently approximate both minimal and almost minimal repairs. The proposed method allows users to substantially reduce the time and effort required for model-based imputation on large datasets without losing accuracy in the final downstream learning task.

**Weaknesses:**

1)It appears the experimental section of this paper only validates the efficacy of the proposed model using MCAR. Why were corresponding experiments not conducted for the previously mentioned MNAR mechanism, as well as the unmentioned MAR mechanism, to provide more sufficient evidence demonstrating the superiority of the proposed method? Furthermore, should the authors consider introducing mixed missing patterns (e.g., some attributes missing via MCAR, others via MNAR) within a single dataset to further validate the method's effectiveness?

2)The abstract mentions a reduction in computational resource consumption, but the experimental section only provides suggestions based on the percentage of imputed samples. It is recommended that the authors also provide corresponding data on the consumption of memory (RAM) and GPU memory (if used). This would more clearly and explicitly demonstrate the reduction in computational resource consumption achieved by the proposed method.


3)I personally believe there is room for further improvement in the comparison algorithms used in the paper. For instance, the algorithms could be categorized into three main groups: statistics-based methods, machine learning-based methods, and deep learning-based methods. For the first two categories, well-established, typical methods can be selected for comparison. For the third category, deep learning-based methods, it is suggested that the authors choose more recent comparison methods. These could include methods based on Variational Autoencoders (VAEs), Generative Adversarial Networks (GANs), and Flow Matching (in addition to the Diffusion-based method already mentioned in the paper, if applicable). I recommend that the author strive to enrich the variety of comparison methods and select the most up-to-date options available.

**Questions:**

See above.

---

> ### Author Response · Authors · 2025-11-26
>
> **W1**\
> We included results for datasets with MAR and MNAR missingness. Due to space limitations in this response, we only report the Malware–MAR results shown in **Table 1**, but the full dataset results (MAR and MNAR) will be included in the revised submission. For the Malware–MAR setting, we did not report **TabCSDI** because the imputation ran for more than two days with no sign of completion. This is due to diffusion-based architecture, which requires repeated forward passes over a very high-dimensional feature space, making the runtime prohibitively long. **MICE** was also infeasible on Malware because its iterative feature-wise regressions construct extremely large design matrices in high dimensions, rapidly exhausting available RAM.
>
> *Table 1. Experimental results of Malware MAR missingness*
> |Data Set|Method  |KNN (Time)     |MICE (Time)      |TSCDI (Time)        |MF (Time)       |KNN (Acc)     |MICE (Acc)|TSCDI (Acc)      |MF (Acc)         |Impute (%)|
> |-|-|-|-|-|-|-|-|-|-|-|
> |Malware20|MR|11.16±0.89|-|27135±2453.1|4726.8±466.2|96.38±0.25|-|96.45±0.13|96.29±0.34|21|
> |Malware20|AC|2.291±0.049|-|5295.3±113.3|4177±6.23|93.39±1.54|-|94.85±0.20|93.97±0.96|3.34±0.15|
> |Malware20|Baseline|17.71|-|-|6830.2|96.30|-|-|96.59|100|
> |Malware40|MR|19.11±1.00|-|65610±3524.5|5422.8±835.9|96.16±0.24|-|96.13±0.06|96.10±0.17|25.23|
> |Malware40|AC|2.178±0.106|-|10534±1322.7|5168.1±90.4|94.55±0.95|-|93.89±0.48|92.14±0.83|3.29±0.22|
> |Malware40|Baseline|36.10|-|-|6460.7|96.15|-|-|95.90|100|
> |Malware60|MR|23.44±0.88|-|76545±5246.8|4224.5±549.4|95.67±0.61|-|95.73±0.10|95.65±1.09|19.69|
> |Malware60|AC|1.807±0.036|-|15259±1631.2|2968.5±25.04|93.44±1.35|-|93.01±0.67|91.78±2.45|3.24±0.23|
> |Malware60|Baseline|48.33|-|-|5794.3|95.85|-|-|96.18|100|
>
> &nbsp; &nbsp; &nbsp; &nbsp; For datasets with original missingness (Table 2), some datasets naturally contain multiple types of missingness, for example, Online-Ed includes both MNAR and MCAR. In the revision, we will clearly describe all missingness types present in the datasets with inherent missing values. We will also follow the reviewer’s recommendation and report results on datasets with injected missingness across multiple mechanisms.
>
> *Table2 Details of datasets with original missing data*
> |Data Set|Task|Features|Training samples|Missing Factor|Missingness Type|
> |-|-|-|-|-|-|
> |Breast Cancer|Classification|10|559|1.97%|MCAR|
> |Water-Pot|Classification|9|2620|39.00%|MCAR|
> |Online-Ed|Classification|36|7026|35.48%|MNAR, MCAR|
> |Bankruptcy|Classification|64|8402|54.00%|MNAR|
> |Air Quality|Regression|12|7344|90.80%|MNAR|
> |Communities|Regression|1954|1595|93.67%|MCAR|
> |Cancer Rate|Regression|32|3048|81.00%|MCAR|
>
> **W2**\
> We reported the reduction in computation time for imputing MR or AMR subsets versus full-data imputation in Table B of the submission. For memory usage, only the Malware–MAR results are available at this stage (shown in the table above), due to space limitations. These partial results already show the pattern: **KNN** consistently uses less RAM under MR than in the baseline, and **MissForest** also shows reduced peak memory across malware20, malware40, and malware60. **MICE** is absent because it was infeasible on this dataset due to memory exhaustion. Overall, the early findings indicate that MR typically lowers memory usage. We will complete the remaining measurements and include full RAM and GPU memory results in the revised submission.
>
> *Table 3. Experimental peak RAM consumption results on Malware with MAR missingness*
> |DataSet|Method|KNN|MICE|RF|
> |-|-|-|-|-|
> |malware20|MR|993.87±126.31|-|2007.92±3.40|
> |malware20|Baseline|994.18|-|2188.73|
> |malware40|MR|917.31±115.00|-|2021.64±5.70|
> |malware40|Baseline|1065.12|-|2187.40|
> |malware60|MR|816.95±163.75|-|1772.30±17.17|
> |malware60|Baseline|1139.50|-|2203.27|
>
> **W3**\
> Our empirical study already covers the three categories noted by the reviewer: statistical methods (**MICE**), traditional ML-based approaches (**KNN** and non-parametric **MissForest**), and deep generative models (**TabCSDI**). **TabCSDI** is a recent diffusion-based method that has been shown to outperform earlier deep-learning imputers such as **GAIN** and **VAE**-based models [*Zheng & Charoenphakdee, NeurIPS 2022*], making it a stronger modern reference than older DL baselines. Likewise, **MICE** remains a widely used and competitive statistical imputation method in practice [*Perini & Nikolic, In Database Data Imputation, SIGMOD 2024*].\
> &nbsp; &nbsp; &nbsp; &nbsp; We agree that adding more deep-learning baselines could strengthen the study. In the revised version, we will include additional models such as **VAE**-, **GAN**-, or flow-based imputers.

---

### Official Review · Reviewer_3w6o · 2025-10-31

**Soundness:** 3
**Presentation:** 2
**Contribution:** 2
**Rating:** 6
**Confidence:** 3

**Summary:**

This paper addresses the high cost of imputing missing data in machine learning. The authors introduce the concepts of Minimal Repair and Almost Minimal Repair, defined as the smallest subsets of missing values that must be imputed to obtain an optimally accurate model or one within a specified error tolerance. They prove that identifying such repairs for support vector machines and linear regression is NP-hard, and propose efficient approximation algorithms with provable guarantees. Extensive experiments show that the proposed methods reduce imputation effort and computational cost while delivering models comparable in accuracy to those trained on fully repaired data.

**Strengths:**

1. The paper introduces the concept of minimal and almost-minimal repair for learning over incomplete data, a problem formulation that offers a new perspective on data preparation by questioning the necessity of full imputation.
2. The work establishes the theoretical complexity of the problem for SVM and linear regression and complements this analysis with the development of practical approximation algorithms accompanied by theoretical error bounds.
3. An extensive empirical evaluation on multiple real-world datasets demonstrates the potential of the proposed methods to reduce the scale of required imputations, suggesting practical utility for resource-constrained learning scenarios.

**Weaknesses:**

1. Section 3.1 notes that the proposed methods rely on the assumption of known domain bounds for each missing value. In practice, such bounds may not be easily determined, and the method for defining them warrants further discussion. The paper could be enhanced by providing specific guidance on how to establish these bounds and by further analyzing how overly wide or narrow bound specifications might impact the algorithm's performance and efficiency.
2. The paper does not seem to address the handling of common categorical variables, for which the concept of "bounds" is not clearly defined.
3. The experimental results indicate that AMR often ​requires more time and yields lower accuracy than​ MR. Therefore, it would be beneficial for the paper to further identify the particular use cases where the trade-off of AMR is most advantageous.
4. The AMR method lacks practical guidance for setting its error threshold e. A sensitivity analysis showing how e impacts both repair size and final model accuracy is needed to demonstrate its practical utility.
5. In Section 6, the paper points out that the methods still perform well empirically even when the theoretical assumptions (such as M-Lipschitz continuity for SVM) are not met. While the results are positive, discussing the potential reasons behind this robustness would help bridge the gap between theory and practice.

**Questions:**

1. The results show that the performance of MR and AMR is largely insensitive to the missingness ratio. Could you please explain the reason behind this?
2. Under what specific conditions would AMR offer a clear advantage in time efficiency over MR, given that it was often slower in the experiments?
3. What practical guidance can be offered for selecting a specific imputation method to use within the MR/AMR framework in a real application?
4. The paper notes the methods work well even when theoretical assumptions are violated. What are the potential reasons for this robustness that could bridge the gap between theory and practice?
5. The provided link to the code repository returns a "file not found" error. Could you please check it?

---

> ### Author Response · Authors · 2025-11-26
>
> **W1**
>
> The feature bounds (min and max values) are often derived from the domain of the feature. When this information is not available, we approximate them using the observed min/max of the feature in the data. We have used this strategy in our experiments, which have resulted in the accurate estimation of MR/AMR and learning accurate models over data with repaired MR/AMR.
>
> **W2**
>
> We handle categorical variables by first converting them into a continuous representation using one-hot encoding, at which point the concept of bounds becomes applicable.
> We have used this method to handle categorical values in the datasets used in the reported empirical studies.
>
> **W3**
>
> According to the results of our empirical study in Table 4, AMR is faster than MR in all classification datasets when imputing with advanced imputation methods, e.g., TabCSDI, and Missing Forest (except for the relatively small Breast Cancer dataset). For regression datasets, AMR is faster than MR in the datasets with the largest number of features (e.g., Communities in Table 6) when using MICE for imputation.
> This is because AMR often needs to impute significantly fewer missing data items than MR.
> This difference becomes larger for computationally intensive imputation methods and large datasets.
> Our algorithm for AMR also returns AMR for all convex models, whereas our algorithms for MR return the desired results only for Linear Regression and SVM.
>
> **W4**
>
> In the final version of the paper, we will conduct the sensitivity analysis of the error threshold.
>
> **W5**
>
> We believe that the empirical robustness of our methods over SVM is due to the stability of the objective function regularization.
> The smooth, strongly convex $L_2$ regularization term of soft-margin SVM may dominate the loss landscape in some cases, effectively reducing the non-smoothness of the Hinge loss.
> Furthermore, because a zero subgradient is the necessary and sufficient condition for global optimality even in the case of non-smoothing loss function, the subgradient norm remains a monotonic proxy for the optimality gap.
> This allows the algorithm to correctly rank and identify critical repairs, even without the strict smoothness assumptions required for the rigorous theoretical bound.
> In the revised submission, we discuss these reasons.
>
> **Q1**
>
> MR and AMR are designed to find an important subset of incomplete samples to impute so that after their imputation, there is no need to impute other data items.
> Therefore, the accuracy of the downstream task is generally insensitive to the missing ratio.
> Our empirical results reported in the submission also show that as the missing ratio of a dataset increases, the MR/AMR algorithms impute a larger ratio of incomplete samples in the dataset.
>
> **Q2**
>
> The algorithm for AMR offers a clear time advantage over MR when the imputation cost is high, as AMR for a dataset is often substantially smaller than its MR. This difference becomes more significant when we use resource-intensive imputation methods, e.g., computationally-intensive methods like TabCSDI or manual imputation, over large datasets. According to our empirical results in Table 4, AMR is faster than MR in all classification datasets when imputing with computationally-intensive methods - TabCSDI, and Missing Forest (except for the relatively small Breast Cancer dataset). For regression datasets, AMR is faster than MR on the dataset with the largest number of features (Communities in Table 6) when imputing using MICE.
>
> **Q3**
>
> The selection of a specific imputation method in our framework follows the same trade-off of cost versus accuracy in picking imputation methods in general: the imputation methods with high accuracy, e.g., manual imputations of complex trained models, often take substantial resources, whereas fast or inexpensive imputation methods may deliver inaccurate results.
> Users often make this decision based on the amount of available resources.
> As our methods reduce the cost and resources needed for imputation, it will enable them to select more methods with high accuracies given a fixed amount of resources.
>
> **Q4**
>
> Please refer to our response to W5.
>
> **Q5**
>
> We have updated the link. We apologize for the confusion. https://anonymous.4open.science/r/Submission_2025-A1C0/README.md

---

### Official Review · Reviewer_21Zj · 2025-11-01

**Soundness:** 2
**Presentation:** 3
**Contribution:** 2
**Rating:** 2
**Confidence:** 4

**Summary:**

The paper proposes the concept of minimal repair (MR) and almost minimal repair (AMR) for learning models over datasets with missing values. The core idea is to identify and impute a minimal subset of missing values that are sufficient to achieve an accurate model, thus reducing the time and computational resources typically required for full imputation. The authors provide theoretical foundations, proving that finding minimal repairs for SVM and linear regression is NP-hard, and propose efficient approximation algorithms with provable error bounds. Through experiments on real-world datasets, the authors demonstrate that their methods (MR and AMR) can reduce imputation time and manual effort without significantly compromising model accuracy.

**Strengths:**

S1. Missing data is a critical issue in real-world datasets, and addressing this problem with minimal imputation is valuable.

S2. The formal definitions of minimal and almost minimal repairs are clear and well-explained.

S3. The paper introduces approximation algorithms with provable error bounds, which is an important step in dealing with the NP-hard problem of finding minimal repairs.

**Weaknesses:**

W1. The core idea of minimal repair has been explored in prior work, especially in ActiveClean and Certain Model Learning, which already aim to reduce the cost of imputation without sacrificing model accuracy. The paper does not provide significant improvements or innovations to justify its claims as groundbreaking.

W2. The experiments are based on a narrow set of models (SVM and linear regression) and imputation methods (KNN, MICE). While these are standard, the results don’t convincingly show why MR/AMR is preferable in practice. The paper would have been stronger with a broader evaluation that includes more recent models or deep learning applications, where missing data issues are also common.

W3. Despite the claims of efficiency, the algorithms for finding MR and AMR can still be computationally expensive, especially in high-dimensional datasets. The claim that MR/AMR will always be faster than full imputation is questionable, particularly when simpler imputation methods like KNN are used. The paper does not sufficiently demonstrate the practical computational benefits for large-scale datasets.

W4. While the NP-hardness results are a nice theoretical contribution, they detract from the practical usability of the method. The algorithms for minimal repair are highly complex, and the paper does not present sufficient evidence to suggest they would be widely applicable in real-world data cleaning scenarios.

W5. The formalism and technical details make the paper hard to follow, especially for readers without a deep background in optimization and data imputation. The dense presentation of the experimental results in the tables, combined with jargon-heavy theoretical discussions, makes the paper less accessible and less impactful.

**Questions:**

Q1. Can the authors show any significant performance improvements when applying MR/AMR to more complex models (e.g., neural networks) or in real-world domains such as healthcare or finance?

Q2. How does the approach scale with high-dimensional datasets (e.g., millions of features or samples)? Can the authors provide additional scalability experiments?

Q3. The paper mentions that the algorithms may be computationally expensive. Could the authors offer further justification for why MR/AMR would be preferred over simpler methods in scenarios where full imputation methods are already cheap to compute?

Q4. What would happen to the performance if the imputation model is poor? Are there any safeguards or adjustments the authors propose to ensure model robustness?

---

> ### Author Response · Authors · 2025-11-28
>
> **W1**
>
> ActiveClean (AC) [*Krishnan et al., ActiveClean: Interactive Data Cleaning For Statistical Modeling, VLDB 2016*] and Certain Model (CM) [*Zhen et al., Certain and Approximately Certain Models for Statistical Learning, SIGMOD 2024*] do not propose the concepts or algorithms of Minimal (MR) or Almost Minimal Repair (AMR).
>
> AC **gradually cleans** subsets deemed relevant to the model in its current SGD iteration. Since early models are often suboptimal, AC often cleans many samples unnecessarily and rarely returns Minimal or Almost Minimal Repairs. Our experiments (e.g., **Table 3**) confirm this empirically.
>
> CM identifies *if* imputation is necessary, but not the minimal subset to repair. We address this gap by strictly defining and finding **minimal** or **almost minimal** subsets.
>
> **W2**
>
> We used representative methods from key groups: statistical (**MICE**), traditional ML-based (**KNN**, **Missing Forest**), and deep generative methods (**TabCSDI**). **TabCSDI** outperforms earlier methods like GAIN [Ref xx], and **MICE** remains a competitive standard in real-world applications [*Perini et al, In-Database Imputation, SIGMOD’24*]. We include other imputation methods in the revised evaluation.
>
> While we proposed MR algorithms only for SVM/LR, our method for **AMR** applies to all convex models. In the revision, we evaluate AMR for other convex models, e.g.,  logistic regression, and outline the path of finding AMR for **DNNs**. Briefly, we leverage the strong representation power of DNNs [*Zhang et al., Understanding deep learning requires
> rethinking generalization, ICLR 2017*], which results in negligible minimum loss across repairs. This simplifies the min-max optimization steps (ST1 and ST2) for DNNs: our algorithm iteratively identifies the subset violating the worst-case threshold (similar to ST2) and updates the model via SGD (similar to ST1), dynamically delivering the ACM.
>
> **W3**
> Our algorithms find AMR very quickly for all datasets. In our empirical studies, we have reported the combined time for finding AMR and imputation of AMR.  We will report them separately in the revised or final versions. Full imputation often fails on large data: for datasets with many features/samples, **MICE** and **TabCSDI** often run out of memory (OOM) or take too long. Using MR/AMR saves the cost of resource-intensive imputation over large data, e.g.,  our algorithm efficiently finds very small AMR subsets (0.24% - 0.73% of samples) for the Malware dataset, MICE runs out of memory (Table B in the appendix). We will use larger datasets in the revised or final versions.
>
> Simple methods like KNN are fast, but they often yield inaccurate models. Applications (e.g., medical) often require accurate imputation using **manual expert verification** or complex models (*Yakout et al., Guided data repair, VLDB 2011*). Because the per-sample cost of these methods is high, full imputation is often infeasible. MR/AMR is critical here as it offers massive savings. For example, in **Table 4**, full imputation with TabCSDI is more than **20x** slower than the total time of finding and imputing the AMR subset.
>
> **W4**
>
> We agree the MR algorithms are complex as they are designed for users prioritizing maximum accuracy. Hence, we developed **AMR**, which offers a relaxed and can be easily implemented using standard SGD.
> We evaluated our methods using real-world datasets from various domains: **Finance** (Credit Default, Bankruptcy), **Healthcare** (Breast Cancer, Cancer Rate), and **Security** (Malware). Results indicate our methods reduce imputation costs while returning accurate models.
>
>
> **W5**
>
> We will revise the paper to improve clarity, make the formalism more accessible, and simplify the presentation of experimental results.
>
> **Q1**
>
> For complex models (DNNs), please refer to **W2**. Regarding real-world domains, please refer to **W4**.
>
> **Q2**
>
> For methods like **TabCSDI** and **Missing Forest**, cost savings from **MR/AMR** become **larger** as data size increases (Tables 4 and 6). On massive datasets, full imputation often becomes intractable—taking days (e.g., Communities in Table 6) or causing OOM errors (e.g., **MICE** on Malware). We will add high-dimensional experiments to demonstrate how **MR/AMR** enables feasible repair where full imputation fails.
>
> **Q3**
>
> Please refer to **W3**.
>
> **Q4**
>
> It is mainly orthogonal to our problem, but we can use our results to address it. We find the range of reliable values imputed data items using the imputation method error rate and use our results, e.g., Theorems 3,12, to report a small set of possible models learned over edge repairs or their average model.

---

### Meta-Review · Area_Chair_Pj3R · 2026-01-05

**Summary:**

This manuscript contributes a approach to learning with incomplete in the logic of "minimal repairs", correcting imputations. The reviews brought forward that the framework is applied to limited baselines, with the choice of supervised learning methods (SVM and linear model) not clearly connected to todays practice. In addition, reviewers found that the theory was hard to follow and did not clearly loop back to practical value of the methods.

**Reviewer Concerns:**

The rebuttal answered the question about limited methods by stating that the results apply to all convex methods, however I do not believe that this answer would convince the reviewers, as most modern methods are not convex.

The reviewers also brought the point that the studied missingness settings were MCAR. The authors addressed this point by adding MNAR experiments which is a welcomed improvement to the work.

The reviewer asked question on the grounding of the methods, ie wether the imputation where converging to true values, to which the authors replied that the purpose was prediction and not true imputation.

**Reviewer Scores:**

I would have expected the reviewers to increase a bit their score, but based on a reading of the paper and the review + rebuttal, I do not believe that this paper would have converged to an acceptance at ICLR, as it is not sufficiently anchored on modern machine learning practices.

---

### Decision · Program_Chairs · 2026-01-26

Reject